# RELATIVE-TRANSLATION INVARIANT WASSERSTEIN DISTANCE

## ABSTRACT

In many real-world applications, data distributions are often subject to translation shifts caused by various factors such as changes in environmental conditions, sensor settings, or shifts in data collection practices. These distribution shifts pose a significant challenge for measuring the similarity between probability distributions, particularly in tasks like domain adaptation or transfer learning. To address this issue, we introduce a new family of distances, relative-translation invariant Wasserstein distances ($RW_p$), to measure the similarity of two probability distributions under distribution shift. Generalizing it from the classical optimal transport model, we show that $RW_p$ distances are also real distance metrics defined on the quotient set $\mathcal{P}_p(\mathbb{R}^n)/\sim$ and invariant to distribution translations, which forms a family of new metric spaces. When $p = 2$, the $RW_2$ distance enjoys more exciting properties, including decomposability of the optimal transport model and translation-invariance of the $RW_2$ distance. Based on these properties, we show that a distribution shift, measured by $W_2$ distance, can be explained in the bias-variance perspective. In addition, we propose two algorithms: one algorithm is a two-stage optimization algorithm for computing the general case of $RW_p$ distance, and the other is a variant of the Sinkhorn algorithm, named $RW_2$ Sinkhorn algorithm, for efficiently calculating $RW_2$ distance, coupling solutions, as well as $W_2$ distance. We also provide the analysis of numerical stability and time complexity for the proposed algorithms. Finally, we validate the $RW_p$ distance metric and the algorithm performance with two experiments. We conduct one numerical validation for the $RW_2$ Sinkhorn algorithm and demonstrate the effectiveness of using $RW_p$ under distribution shift for similar thunderstorm detection. The experimental results report that our proposed algorithm significantly improves the computational efficiency of Sinkhorn in practical applications, and the $RW_p$ distance is robust to distribution translations.

## 1 INTRODUCTION

Optimal transport (OT) theory and Wasserstein distance (Peyré & Cuturi, 2020; Janati et al., 2020a; Villani, 2009) provide a rigorous measurement of similarity between two probability distributions. Numerous state-of-the-art machine learning applications are developed based on the OT formulation and Wasserstein distances, including domain adaptation, score-based generative model, Wasserstein generative adversarial networks, Fréchet inception distance (FID) score, Wasserstein auto-encoders, distributionally robust Markov decision processes, distributionally robust regressions, graph neural networks based objects tracking, etc (Shen et al., 2017; Pinheiro, 2017; Courty et al., 2017b; Damodaran et al., 2018; Courty et al., 2017a; Arjovsky et al., 2017; Heusel et al., 2017; Tolstikhin et al., 2017; Clement & Kroer, 2021; Shafieezadeh-Abadeh et al., 2015; Chen & Paschalidis, 2018; Yu et al., 2023; Sarlin et al., 2019). However, the classical Wasserstein distance has major limitations in certain machine learning and computer vision applications. For example, a meteorologist often focuses on identifying similar weather patterns in a large-scale geographical region Wang et al. (2023); Roberts & Lean (2008); Dixon & Wiener (1993), where he/she cares more about the "shapes" of weather events rather than their exact locations. The weather events are represented as images or point clouds from the radar reflectivity map. Here the classical Wasserstein distance is not useful since the relative location difference or relative translation between two very similar weather patterns will add to the Wasserstein distance value. Another example is the inevitable distribution shift in real-world

datasets. A distribution shift may be introduced by sensor calibration error, environment changes between train and test datasets, simulation to real-world (*sim2real*) deployment, etc. Motivated by these practical use cases and the limitations of Wasserstein distances, we ask the following research question:

*Can we find a new distance metric and a corresponding efficient algorithm to measure the similarity between probability distributions (and their supports) regardless of their relative translation?*

To answer this research question, we introduce the relative translation optimal transport (ROT) problem and the corresponding relative-translation invariant Wasserstein distance $RW_p$. We then focus on the general case result when $p \in [1, \infty)$ and the quadratic case ($p = 2$) by identifying two exciting properties of the $RW_2$ distance. We leverage these properties to design a variant of the Sinkhorn algorithm to compute $RW_2$ distance, coupling solutions, as well as $W_2$ distance. In addition, we provide analysis and numerical experiment results to demonstrate the effectiveness of the new $RW_2$ distance against translation shifts. Finally, we show the scalability and practical usage of the $RW_2$ in a real-world meteorological application.

**Contributions.** The main contributions of this paper are highlighted as follows: *(a)* we introduce a family of new similarity metrics, relative-translation invariant Wasserstein ($RW_p$) distances, which are real distance metrics like the Wasserstein distance and invariant to the relative translation of two distributions; *(b)* we identify two useful properties of the quadratic case $RW_2$ to support our algorithm design: decomposability of the ROT problem and translation-invariance of both the ROT problem solution and the resulted $RW_2$; *(c)* we show the non-convexity of general ROT problem and propose a two-stage algorithm for computing the general $RW_p$ distances; and *(d)* we propose an efficient variant of Sinkhorn algorithm, named the $RW_2$ Sinkhorn, for calculating $RW_2$ distance, coupling solutions as well as $W_2$ distance with significantly reduced computational complexity and enhanced numerical stability. Empirically, we report promising performance from the proposed $RW_2$ distance when the relative translation is large, and the $RW_2$ Sinkhorn algorithm in illustrative numerical examples and a large-scale real-world task for similar weather detection. Figure 1 shows our major findings in this work.

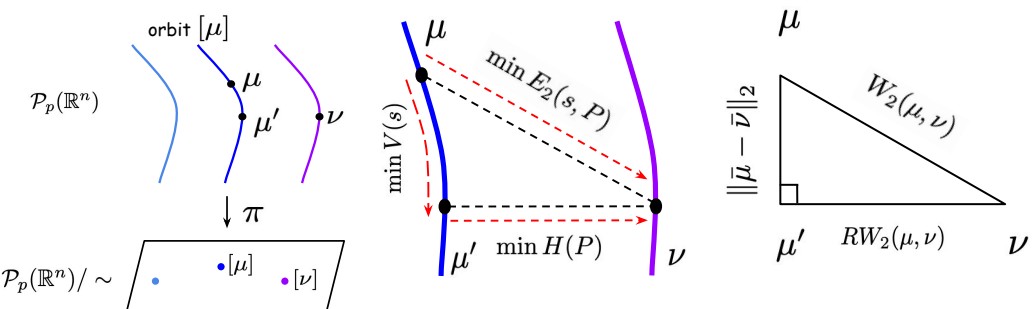

(a) Schematic illustration of the quotient set $\mathcal{P}_p(\mathbb{R}^n)/ \sim$, where $\pi$ stands for the natural projection from $\mathcal{P}_p(\mathbb{R}^n)$ to $\mathcal{P}_p(\mathbb{R}^n)/ \sim$ induced by the translation relation. The equivalence class (orbit) $[\mu]$ is pictured as the blue line of $\mu$ and $\mu'$ in $\mathcal{P}_p(\mathbb{R}^n)$ and it corresponds to a point $[\mu]$ in the quotient set $\mathcal{P}_p(\mathbb{R}^n)/ \sim$.

(b) Decomposition of the optimal transport optimization. To move $\mu$ to $\nu$, $\mu$ can be moved along the orbit (equivalence class) $[\mu]$ to $\mu'$ first, which is related to the vertical optimization $V(s)$, then moved on the quotient set $\mathcal{P}_2(\mathbb{R}^n)/ \sim$ to the target $\nu$, which is related to the horizontal optimization $H(P)$.

(c) Pythagorean relationship of the distances. Three types of distances, $W_2$, $RW_2$ and $\|\bar{\mu} - \bar{\nu}\|_2$, are used to measure the minimal values of the three objective functions, $E_2(P, s)$, $H(P)$ and $V(s)$, respectively, as shown in the subfigure (b).

Figure 1: The relative translation optimal transport problem and $RW_p$ distances.

**Notations.** Let $\mathcal{P}_p(\mathbb{R}^n)$ be the set of all probability distributions with *finite* moments of order $p$ defined on the space $\mathbb{R}^n$. For simplicity, we assume $\mu$ and $\nu$ represent a pair of source and target distributions, respectively. Assume that $m_1$ and $m_2$ are the number of supports when distribution $\mu$ and $\nu$ have finite supports $\{x_i\}_{i=1}^{m_1}$ and $\{y_j\}_{j=1}^{m_2}$. Let $\mathbb{R}_*^{m_1 \times m_2}$ represents the set of all $m_1 \times m_2$ matrices with non-negative entries. $[\mu]$ represents the equivalence class (orbit) of $\mu$ under the shift equivalence relation in $\mathcal{P}_p(\mathbb{R}^n)$. $\bar{\mu}$ and $\bar{\nu}$ represents the mean of probability distribution $\mu$

and $\nu$, respectively. $\mathbf{e}_m$ denotes a vector in $\mathbb{R}^m$ where all elements are ones. $./$ represents the component-wise vector division.

**Related work.** Optimal transport theory is a classical area of mathematics with strong connections to probability theory, diffusion processes and PDEs. Due to the vast literature, we refer readers to (Villani & Society, 2003; Ambrosio et al., 2005; Villani, 2009; Oll, 2014) for comprehensive reviews. Computational OT methods have been widely explored, including Greenkhorn algorithm (Altschuler et al., 2017), Network Simplex method (Peyré & Cuturi, 2020), Wasserstein gradient flow (Mokrov et al., 2021; Fan et al., 2022), neural network approximation (Chen & Wang, 2023). Significant research has also been conducted on Wasserstein distances, such as the sliced Wasserstein distance (Nguyen & Ho, 2023; Mahey et al., 2023; Nguyen & Ho, 2022), Gromov-Wasserstein distance (Sejourne et al., 2021; Le et al., 2022; Alvarez-Melis et al., 2019), etc. Other important topics include Wasserstein barycenter (Guo et al., 2020; Vaskevicius & Chizat, 2023; Korotin et al., 2022; Lin et al., 2020; Korotin et al., 2021) and unbalanced optimal transport (Nguyen et al., 2024; Chizat, 2017).

Among these foundational areas, information geometry (Amari, 2016; Liero et al., 2018; Janati et al., 2020b) and the Wasserstein-Bures metric (Chen et al., 2015; Bhatia et al., 2019; Peyré & Cuturi, 2020; Malagò et al., 2018) are closely related to our work, as both provide tools for measuring variances. However, it is important to note key differences. Unlike information geometry, which typically employs measures such as Bregman divergence or statistical information, our approach utilizes the energy transport cost as the primary metric. Additionally, while the Wasserstein-Bures metric specifically focuses on Gaussian distributions and the $W_2$ metric, our research extends to more general distributions and considers broader classes of $p$-norm metrics, offering a more comprehensive framework for analysis.

## 2 PRELIMINARIES

Before delving into the details of our proposed method, it is essential to focus on the groundwork with an introduction to key aspects of classical optimal transport theory and formulations. This foundation will support the subsequent derivations and proofs presented in Section 3.

### 2.1 OPTIMAL TRANSPORT THEORY

The optimal transport theory focuses on finding the minimal-cost transport plans for moving one probability distribution to another probability distribution in a metric space. The core of this theory involves a cost function, denoted as $c(x, y)$, alongside two probability distributions, $\mu(x)$ and $\nu(y)$. The optimal transport problem is to find the transport plans (coupling solutions) that minimize the cost of moving the distribution $\mu(x)$ to $\nu(y)$, under the cost function $c(x, y)$. Although the cost function can take any non-negative form, our focus will be on those derived from the $p$-norm, expressed as $\|x - y\|_p^p$ for $p \in [1, \infty)$, since the optimal transport problem is well-defined (Villani, 2009).

Assuming $\mu(x)$ as the source distribution and $\nu(y)$ as the target distribution, $\mu, \nu \in \mathcal{P}_p(\mathbb{R}^n)$, we can formulate the optimal transport problem as a functional optimization problem, detailed below:

**Definition 1** ($p$-norm optimal transport problem (Villani, 2009)).

$$OT(\mu, \nu, p) = \min_{\gamma \in \Gamma(\mu, \nu)} \int_{\mathbb{R}^{2n}} \|x - y\|_p^p d\gamma(x, y), \tag{1}$$

*with* $\Gamma(\mu, \nu) = \{\gamma \in \mathcal{P}_p(\mathbb{R}^{2n}) | \int_{\mathbb{R}^n} \gamma(x, y) dx = \nu(y), \ \int_{\mathbb{R}^n} \gamma(x, y) dy = \mu(x), \ \gamma(x, y) \geq 0\}$.

Here $\gamma(x, y)$ represents the transport plan (or the coupling solution), indicating the amount of probability mass transported from source support $x$ to target support $y$. The objective function is to minimize the total transport cost, which is the integrated product cost of distance and transported mass across all source-target pairs $(x, y)$.

After the foundational optimal transport problem is outlined, we can introduce a family of real metrics, the Wasserstein distances, for measuring the distance between probability distributions on the set $\mathcal{P}_p(\mathbb{R}^n)$. These distances are defined based on the optimal transport problem.

**Definition 2** (Wasserstein distances (Villani, 2009)). *The Wasserstein distance between $\mu$ and $\nu$ is the $p$th root of the minimal total transport cost from $\mu$ to $\nu$, denoted as $W_p, p \in [1, \infty)$:*

$$W_p(\mu, \nu) = OT(\mu, \nu, p)^{\frac{1}{p}}. \tag{2}$$

The Wasserstein distance is a powerful tool for assessing the similarity between probability distributions. It is a real metric admitting the properties of indiscernibility, non-negativity, symmetry, and triangle inequality Villani (2009). Meanwhile, it is well-defined for any probability distribution pairs, including discrete-discrete, discrete-continuous, and continuous-continuous.

For practical machine learning applications, the functional optimization described in Equation (1) can be adapted into a discrete optimization framework. This adaptation involves considering the distributions, $\mu$ and $\nu$, as comprised of *finite* supports, $\{x_i\}_{i=1}^{m_1}$ and $\{y_j\}_{j=1}^{m_2}$, with corresponding probability masses $\{a_i\}_{i=1}^{m_1}$ and $\{b_j\}_{j=1}^{m_2}$, respectively, where $m_1$ and $m_2$ are the number of supports (data points). Since all $m_1$ and $m_2$ are finite numbers, we can use an $m_1 \times m_2$ matrix $C$ to represent the cost between supports, where each entry represents the transporting cost from $x_i$ to $y_j$, i.e., $C_{ij} = \|x_i - y_j\|_p^p$. This discrete version of the optimal transport problem can then be expressed as a linear programming problem, denoted as $\text{OT}(\mu, \nu, p)$:

$$\text{OT}(\mu, \nu, p) = \min_{P \in \Pi(\mu, \nu)} \sum_{i=1}^{m_1} \sum_{j=1}^{m_2} P_{ij} C_{ij}, \tag{3}$$

with $\Pi(\mu, \nu) = \{P \in \mathbb{R}_*^{m_1 \times m_2} | P \, \mathbf{e}_{m_1} = a, P^\top \, \mathbf{e}_{m_2} = b\}$, where $\Pi(\mu, \nu)$ is the feasible set of this problem, vectors $a$ and $b$ are the probability masses of $\mu$ and $\nu$, respectively. coupling solutions $P_{ij}$ indicates the amount of probability mass transported from the source point $x_i$ to the target point $y_j$. This linear programming approach provides a scalable and efficient way for solving discrete optimal transport problems in various data-driven applications.

## 2.2 Sinkhorn Algorithm

Equation (3) formulates a linear programming problem, which is commonly solved by simplex methods or interior-point methods Peyré & Cuturi (2020). Because of the special structure of the feasible set $\Pi(\mu, \nu)$, another approach for solving this problem is to transform it into a matrix scaling problem by adding an entropy regularization in the objective function Cuturi (2013). The matrix scaling problem can be solved by the Sinkhorn algorithm, which is an iterative algorithm that enjoys both efficiency and scalability. In detail, the Sinkhorn algorithm will initially assign $u^{(0)}$ and $v^{(0)}$ with vector $\mathbf{e}_{m_1}$ and $\mathbf{e}_{m_2}$, then the vector $u^{(k)}$ and $v^{(k)}$ ($k \geq 1$) are updated alternatively by the following equations:

$$u^{(k+1)} \leftarrow a./Kv^{(k)}, \quad v^{(k+1)} \leftarrow b./K^\top u^{(k+1)}, \tag{4}$$

where $K_{ij} = e^{-\frac{C_{ij}}{\lambda}}$ ($\lambda$ is the coefficient of the entropy regularized term) and the division is component-wise. When the convergence precision is satisfied, the coupling solution $P$ will be calculated by the matrix diag(u)$K$diag(v). It has been proved the solution calculated by the Sinkhorn algorithm can converge to the exact coupling solution of the linear programming model, as $\lambda$ goes to zero (Cominetti & Martín, 1994). One caveat of this calculation is the exponent operation, which may cause "division by zero", we will show how we can improve the numerical stability in Section 4.

# 3 Relative Translation Optimal Transport and $RW_p$ Distances

Here we present the relative translation optimal transport model and the $RW_p$ distances. We first introduce the theoretical understanding of the relative translation optimal transport problem and the $RW_p$ distances. We will then focus on computational tractability on those $RW_p$ distances. Finally, we focus on the quadratic case ($RW_2$) and its properties. For simplicity, we present the results for discrete distributions; however, because of the weak convergence property of Wasserstein distances, these results are also applicable to arbitrary distributions in set $\mathcal{P}_p(\mathbb{R}^n)$.

## 3.1 Relative Translation Optimal Transport Formulation and $RW_p$ Distances

As discussed in Section 1, the classical optimal transport (OT) problem is not very precise to the case when there is a relative translation allowed between two distributions (or the two datasets known as their supports). We introduce the *relative translation optimal transport* problem, $ROT(\mu, \nu, p)$, which is formulated to find the minimal total transport cost under any translation.

**Definition 3** (Relative translation optimal transport problem). *Continuing with the previous notations,*

$$ROT(\mu, \nu, p) = \inf_{s \in \mathbb{R}^n} \min_{P \in \Pi(\mu, \nu)} E_p(s, P), \tag{5}$$

*where variable $s$ represents the translation of source distribution $\mu$, variables $P_{ij}$ represent the coupling solution between the support $x_i$ and the support $y_j$, and $E_p(s, P)$ represents the total transport cost under $p$ norm, i.e. $E_p(s, P) = \sum_{i=1}^{m_1} \sum_{j=1}^{m_2} P_{ij} \|x_i - y_j + s\|_p^p$.*

The ROT problem can be viewed as a generalized form of the classical OT in Equation (1). There are two stages in this optimization. The inner stage is exactly the classical OT, whereas the outer stage finds the optimal relative translation for the source distribution to minimize the total transport cost.

**Theorem 1** (Compactness and existence of the minimizer)**.** *For Equation* (5)*, the domain of the variable $s$ can be restricted on a compact set $\Omega = \{s \in \mathbb{R}^n | \|s\|_p \leq 2 \max_{ij} \|x_i - y_j\|_p\}$. Thus, we have*

$$ROT(\mu, \nu, p) = \min_{s \in \mathbb{R}^n} \min_{P \in \Pi(\mu, \nu)} E_p(s, P),$$

*where the minimum can be achieved.*

The proof of Theorem 1 is provided in Appendix A.

From the perspective of equivalence relation, we could have a better view of which space the ROT problem is defined on. Assume that $\sim$ is the translation relation on the set $\mathcal{P}_p(\mathbb{R}^n)$. When distribution $\mu$ can be translated to distribution $\mu'$, we denote it by $\mu \sim \mu'$. Because the translation is an equivalence relation defined on the set $\mathcal{P}_p(\mathbb{R}^n)$, we may partition set $\mathcal{P}_p(\mathbb{R}^n)$ by the translation relation, which leads to a quotient set, $\mathcal{P}_p(\mathbb{R}^n)/\sim$. $\mathcal{P}_p(\mathbb{R}^n)/\sim$ consists of the equivalence class of distributions, and each equivalence class, denoted by $[\mu]$, contains all mutually translatable probability distributions. Therefore, the ROT problem can also be regarded as an OT problem defined on the quotient set, $\mathcal{P}_p(\mathbb{R}^n)/\sim$, which tries to find the minimal total transport cost between $[\mu]$ and $[\nu]$. Figure 1(a) illustrates this idea. We can see that the value of the ROT problem is invariant to translations of either source or target distributions.

Building upon the ROT model, we introduce a new family of Wasserstein distances to measure the minimal total transport cost between different equivalence classes of probability distributions. As mentioned above, the value of the ROT problem is invariant to any relative translations, thus, we name the corresponding Wasserstein distances as relative-translation invariant Wasserstein distances, denoted by $RW_p$:

**Definition 4** (Relative-translation invariant Wasserstein distances)**.**

$$RW_p(\mu, \nu) = ROT(\mu, \nu, p)^{\frac{1}{p}}.$$

Similar to the situation where $W_p$ is a real metric on $\mathcal{P}_p(\mathbb{R}^n)$, we can obtain the following theorem.

**Theorem 2.** *$RW_p$ is a real metric on the quotient set $\mathcal{P}_p(\mathbb{R}^n)/\sim$.*

The proof of Theorem 2 is provided in Appendix A. It should be noted that we would not take "relative rotation" into account in our equation 5, since relative rotation will violate the metric properties.

### 3.2 $RW_p$ METRIC SPACES

One advantage of this family of distances is that it defines a new family of metric spaces $(\mathcal{P}_p(\mathbb{R}^n)/\sim, RW_p)$. These spaces differ from the conventional metric spaces $(\mathcal{P}_p(\mathbb{R}^n), W_p)$ (Villani, 2009), as the distances here are solely influenced by the "shape" of the variances, independent of their means.

Classical $L_p$ models show that the $L_1$ norm exhibits enhanced robustness to outliers, making it more appropriate for noisy data applications (Jolliffe, 2002; Zou et al., 2004). In contrast, the $L_2$ norm does not induce sparsity, thereby reducing its effectiveness in feature selection. Similarly, $RW_1$ distance is anticipated to offer greater robustness in the presence of noise, whereas $RW_2$ distance is expected to perform more balanced in cleaner datasets.

### 3.3 COMPUTATIONAL TRACEABILITY OF $RW_p$

When the problem is defined in one-dimensional space, it is straightforward to confirm that the ROT problem is convex w.r.t. the variable $s$ for any $p \in [1, \infty)$, due to the monotonic behavior of their cumulative distribution functions.

In high-dimensional space, the original ROT problem is no longer consistently computationally tractable as in one dimension. Some counterexamples reveal that the outer function $\min_{P \in \Pi(\mu, \nu)} E_p(s, P)$

is non-convex w.r.t. the variable $s$. In addition, we also consider two related reformulated problems, $\min_{P \in \Pi(\mu,\nu)} \min_{s \in \mathbb{R}^n} E_p(s,P)$ and $\min_{(s,P)} E_p(s,P)$, and several counterexamples also show both function $\min_{s \in \mathbb{R}^n} E_p(s,P)$ and $\min_{(s,P) \in \Omega} E_p(s,P)$ are non-convex w.r.t. variable $P$ and variable $(P,s)$, respectively. (All counterexamples as mentioned above are provided in Appendix C).

**Theorem 3** (Closed-form gradient). *For the optimization problem* $\min_{P \in \Pi(\mu,\nu)} \min_{s \in \mathbb{R}^n} E_p(s,P)$, *denoting the outer function* $\min_{s \in \mathbb{R}^n} E_p(s,P)$ *by* $F_p(P)$ , *we have:*

$$\nabla_P F_p(P) = C(s_P),$$

*where* $C_{ij}(s) = \|x_i + s - y_j\|_p^p$ *and* $s_P$ *satisfies with constraint* $\sum_{i=1}^{m_1} \sum_{j=1}^{m_2} P_{ij} \operatorname{sign}(x_i + s_P - y_j)\|x_i + s_P - y_j\|_p^{p-1} = 0$.

The proof of Theorem 3 is provided in Appendix A. Based on the closed-form of the gradient of $F_p(P)$ in Theorem 3, we design our algorithms to compute $RW_p$ distances in Section 4.

### 3.4 QUADRATIC ROT AND PROPERTIES OF THE $RW_2$ DISTANCE

We show two useful properties in the quadratic case of ROT and the resulted $RW_2$ distance: decomposability of the ROT optimization model (Theorem 4), translation-invariance of coupling solutions of the ROT problem (Corollary 1).

**Theorem 4** (Decomposition of the quadratic ROT). *The two-stage optimization problem in quadratic ROT can be decomposed into two independent single-stage optimization problems:*

$$ROT(\mu,\nu,2) = \min_{s \in \mathbb{R}^n} \min_{P \in \Pi(\mu,\nu)} E_2(s,P) = \min_{P \in \Pi(\mu,\nu)} H(P) + \min_{s \in \mathbb{R}^n} V(s) \tag{6}$$

*where horizontal function* $H(P) = \sum_{i=1}^{m_1} \sum_{j=1}^{m_2} P_{ij}\|x_i - y_j\|_2^2$ *and vertical function* $V(s) = \|s\|_2^2 + 2s \cdot (\bar{\mu} - \bar{\nu})$.

Function $E_2(s,P)$, $H(P)$ and $V(s)$ are illustrated in Figure 1(b). The proof of Theorem 4 is provided in Appendix A.

Theorem 4 is the core idea for the $RW_2$ algorithm design in Section 4. It indicates that the coupling solutions $P$ to the OT problem are always the same as its ROT version, and verse versa, i.e.,

**Corollary 1** (Translation-invariance of both the ROT solution and $RW_2$). *The coupling solutions to the quadratic ROT problem are invariant to any translation of distributions.*

Corollary 1 not only guarantees the robustness of $RW_2$ against translational shifts but also suggests that the coupling solution of an ROT problem (including the classical OT problem) can be calculated by a "more stable" cost matrix. This helps us improve the numerical stability and reduce the time complexity in many practical conditions. We provide a detailed analysis in Section 4 and demonstrate it in Section 5.

**Corollary 2** (Relationship between $RW_2$ and $W_2$). *Let $s$ be the minimizer* $\bar{\nu} - \bar{\mu}$, *it follows that,*

$$W_2^2(\mu,\nu) = \|\bar{\mu} - \bar{\nu}\|_2^2 + RW_2^2(\mu,\nu). \tag{7}$$

Corollary 2 indicates that there exists a Pythagorean relationship among three types of distances, $W_2$, $RW_2$, and $L_2$, as illustrated in Figure 1(c). This relationship extends the Wasserstein-Bures metric (Chen et al., 2015; Bhatia et al., 2019; Peyré & Cuturi, 2020; Malagò et al., 2018), which applies specifically to Gaussian distributions.

Corollary 2 provides a refinement to understand a distribution shift (measured by $W_2$) from bias and variance decomposition. The $L_2$ Euclidean distance between the expectations of two distributions corresponds to the "bias" between two distributions, and the value of $RW_2$ corresponds to the difference of "variances" or the "shapes" of two distributions.

## 4 $RW_p$ ALGORITHMS AND $RW_2$ TECHNIQUE

### 4.1 $RW_p$ ALGORITHMS

Based on the Theorem 3, we propose $RW_p$ algorithms ($p \geq 1$) to compute the general $RW_p$ distances by updating variable $P$ and $s$ alternatively, as shown in Algorithm 1. Note that, when $p = 1$, we

can also incorporate Proximal gradient descent (Moreau Envelope) to reduce the non-smooth of $\nabla_t E_p(t, P)$. When $p = 2$, we can take advantage of the Thoorem 4 to speed up.

---

**Algorithm 1** $RW_p$ Algorithms

---

1: **Input:** $\{x_i\}_{i=1}^{m_1}, \{y_j\}_{j=1}^{m_2}, \{a_i\}_{i=1}^{m_1}, \{b_j\}_{i=1}^{m_2}, p, \epsilon_1, \epsilon_2, \eta_1, \eta_2.$
2: **Output:** The value of $RW_p$ distance.
3: $P^{(0)} \leftarrow a \cdot b^\top, s^{(0)} \leftarrow 0, C^{(0)} \leftarrow 0, k \leftarrow 0$
4: **repeat**
5:    **repeat**
6:       **for** $i = 1$ **to** $m_1$ **do**
7:          **for** $j = 1$ **to** $m_2$ **do**
8:             $f_{ij} \leftarrow \text{sign}(x_i - y_j + t^{(l)})\|x_i - y_j + t^{(l)}\|_p^{p-1}$
9:       $\nabla_t E_p(t, P) \leftarrow \sum_{i=1}^{m_1} \sum_{j=1}^{m_2} P_{ij} f_{ij}$
10:       $t^{(l+1)} \leftarrow t^{(l)} - \eta_1 \nabla_t E_p(t, P)$
11:       $l \leftarrow l + 1$
12:    **until** $\|\nabla_t E_p(t, P)\|_p^p \le \epsilon_1$
13:    $s^{(k+1)} = t^{(l)}$
14:    **for** $i = 1$ **to** $m_1$ **do**
15:       **for** $j = 1$ **to** $m_2$ **do**
16:          $C_{ij} \leftarrow \|x_i + s^{(k+1)} - y_j\|_p^p$
17:    $P^{(k+1)} \leftarrow \underset{P}{\arg\min} \, OT(a, b, C, P)$
18:    $k \leftarrow k + 1$
19: **until** $\|s^{(k)} - s^{(k-1)}\|_p^p \le \epsilon_2$
20: **return** $(\sum_{i=1}^{m_1} \sum_{j=1}^{m_2} C_{ij}^{(k)} P_{ij}^{(k)})^{\frac{1}{p}}$

---

where $\underset{P}{\arg\min} \, OT(a, b, C, P)$ can be solved by the Sinkhorn algorithm or LP solvers.

## 4.2 $RW_2$ Algorithm

Based on Theorem 4 and Corollary 1, we propose the $RW_2$ Sinkhorn algorithm for computing $RW_2$ distance and coupling solution $P$, which is described in Algorithm 2. The key idea of this algorithm involves precomputing the difference between the means of two distributions, as shown in Line 3. Subsequently, it addresses a specific instance of the optimal transport problem where the means of the two distributions are identical by a regular Sinkhorn algorithm. It is important to note that alternative algorithms, such as the network-simplex algorithm or the auction algorithm (Peyré & Cuturi, 2020), can also be employed to complete the specific instance procedure.

---

**Algorithm 2** $RW_2$ Sinkhorn Algorithm

---

1: **Input:** $\{x_i\}_{i=1}^{m_1}, \{y_j\}_{j=1}^{m_2}, \{a_i\}_{i=1}^{m_1}, \{b_j\}_{i=1}^{m_1}, \lambda, \epsilon.$
2: **Output:** $RW_2, P.$
3: $s \leftarrow \sum_{j=1}^{m_2} y_j b_j - \sum_{i=1}^{m_1} x_i a_i$
4: **for** $i = 1$ **to** $m_1$ **do**
5:    **for** $j = 1$ **to** $m_2$ **do**
6:       $C_{ij} \leftarrow \|x_i + s - y_j\|_2^2$
7: $K \leftarrow \exp(-C/\lambda)$
8: $u^{(0)} \leftarrow e_{m_1}, \quad v^{(0)} \leftarrow e_{m_2}, \quad k \leftarrow 0$
9: **repeat**
10:    $u^{(k+1)} \leftarrow a./(Kv^{(k)})$
11:    $v^{(k+1)} \leftarrow b./(K^\top u^{(k+1)})$
12:    $P \leftarrow \text{diag}(u^{(k+1)})K\text{diag}(v^{(k+1)})$
13:    $k \leftarrow k + 1$
14: **until** $\|Pe_{m_1} - a\|_2^2 + \|P^\top e_{m_2} - b\|_2^2 \le \epsilon$
15: **return:** $\sum_{i=1}^{m_1} \sum_{j=1}^{m_2} P_{ij} C_{ij}, P$

---

## 4.3 $RW_2$ Technique for $W_2$ Computation

With the observation of Corollary 2, we can propose a new improvement to compute the $W_2$ distance from the right side of the Equation (7). When $\|\bar{\nu} - \bar{\mu}\|_2$ is large enough, this improvement performs better than the original Sinkhorn in terms of numerical stability and time complexity. We analyze this new approach in the rest of this section. In addition, the experiment in Section 5.1 validates the analysis of our proposed $RW_2$ Sinkhorn algorithm for computing $W_2$ distance.

## 4.4 Numerical Stability and Complexity Analysis

The division by zero is a common numerical issue of the Sinkhorn algorithm (Peyré & Cuturi, 2020). As shown in Equation (4), infinitesimal value often occurs in the exponential process of the (negative)

cost matrix, $K \leftarrow e^{-\frac{C}{\lambda}}$. The results of the Corollary 1 suggest that it is possible to switch to another "mutually translated" cost matrix under a relative translation $s$ to increase the numerical stability while preserving the same optimal solutions.

To measure the numerical stability of a matrix, we introduce $g(K)$, defined by the product of all entries. As $g(K)$ increases, most entries $K_{ij}$ deviate from zero, which means numerical computation will be more stable. Since $g(K) = \prod_{i=1}^{m_1} \prod_{j=1}^{m_2} K_{ij} = \prod_{i=1}^{m_1} \prod_{j=1}^{m_2} \exp\left(-\frac{C_{ij}}{\lambda}\right) = \exp\left(-\frac{\sum_{i=1}^{m_1} \sum_{j=1}^{m_2} \|x_i + s - y_j\|_2^2}{\lambda}\right)$, one can verify the maximizer of $g(K)$ is when the relative translation $s = \bar{y} - \bar{x}$, which is almost equal to $\bar{\nu} - \bar{\mu}$ when the probability mass of the samples is the same.

Altschuler et al. (2017) shows that the time complexity of the optimal transport model by Sinkhorn algorithm with $\tau$ approximation is $O(m^2\|C\|_\infty^3 (\log m)\tau^{-3})$, where $\|C\|_\infty = \max_{ij} C_{ij}$ and assuming $m = m_1 = m_2$ for the sake of simplicity. The following theorem indicates that for a wide range of distributions, the translated cost matrix has a smaller infinity norm $\|C\|_\infty$. Thus, the time complexity of the algorithm will be reduced.

**Theorem 5.** *Let $\mu, \nu$ be two high-dimensional sub-Gaussian distributions in $\mathbb{R}^n$. $(X_1, X_2, \ldots, X_{m_1})$, $(Y_1, Y_2, \ldots, Y_{m_2})$ are i.i.d data sampled from $\mu$ and $\nu$ separately. Let $\bar{\mu} = \mathbb{E}\mu$, $\bar{\nu} = \mathbb{E}\nu$, $\bar{X} = \sum_{i=1}^{m_1} X_i/m_1$, $\bar{Y} = \sum_{i=1}^{m_2} Y_i/m_2$. Assume $\|\mu - \bar{\mu}\|_{\psi_2} < \infty$, $\|\nu - \bar{\nu}\|_{\psi_2} < \infty$. Let $l = \|\bar{\mu} - \bar{\nu}\|_2$ be the distance between the centers of the two distributions. If it satisfies:*

$$l \geq L\sqrt{n}\left[1 + \|\mu - \bar{\mu}\|_{\psi_2} + \|\nu - \bar{\nu}\|_{\psi_2}\right]$$

$$+ L\left[\sqrt{\log(4m_1/\delta)} \cdot \|\mu - \bar{\mu}\|_{\psi_2} + \sqrt{\log(4m_2/\delta)} \cdot \|\nu - \bar{\nu}\|_{\psi_2}\right],$$

*where $L$ is an absolute constant, then with probability at least $1 - \delta$, we have*

$$\max_{i,j} \|X_i - \bar{X} - Y_j + \bar{Y}\|_2 \leq \max_{i,j} \|X_i - Y_j\|_2.$$

**Remark 1.** *Sub-Gaussian distributions represent a broad class of distributions that encompass many common types, including multivariate normal distribution, multivariate symmetric Bernoulli, and uniform distribution on the sphere. Theorem 5 demonstrates that when the distance between the centers of the two distributions is significantly large, the maximum absolute entry of the cost matrix $\|C\|_\infty = \max_{ij} |C_{ij}|$ after translation tends to decrease. Consequently, our $RW_2$ method achieves better time complexity compared to $W_2$. This theoretical finding is consistent with our experimental results, as shown in Figure 3. Detailed proof about Theorem 5 will be postponed to Appendix B.*

## 5 EXPERIMENTS

To evaluate our proposed methods, we conducted two experiments: numerical validation and weather pattern detection. The first one validates the computational time and error of the $RW_2$ Sinkhorn algorithm and the second one demonstrates the scalability of $RW_2$ and $RW_p$ for identifying similar weather patterns in large datasets. Both experiments were run on a 2.60 GHz Intel Core i7 processor with 16GB RAM.

### 5.1 NUMERICAL VALIDATION

We first demonstrate the advantages of using the $RW_2$ Sinkhorn algorithm to compute $W_2$ distance with specially designed examples. Two data sets, $\mu$ and $\nu$, each containing 1,000 samples, are drawn from identical distributions. To compare Algorithm 2 with the original Sinkhorn, we slightly translate $\mu$ by a vector $s$, with translation lengths ranging from $[0, 3]$, as illustrated in Figure 2.

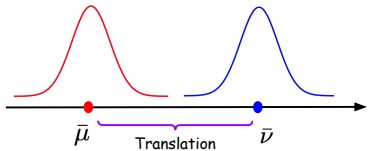

Figure 2: Schematic of the first experiment: two sample sets, $\mu$ and $\nu$, are drawn from the same distribution. To evaluate the performance of the $RW_2$ algorithm versus the original Sinkhorn, we translate $\mu$ by the vector $s = \bar{\nu} - \bar{\mu}$.

**Settings** We compare two versions of the Sinkhorn algorithms in $W_2$ error and running time, repeating each experiment 10 times. We evaluate Gaussian distributions in $\mathbb{R}$ (Figure 3(a) and 3(b)) and in $\mathbb{R}^{10}$ (Figure 3(c) and 3(d)). For both algorithms, we set $\lambda = 0.1$ and $\epsilon = 1 \times 10^{-9}$ calling **ot.sinkhorn2()** function from Python optimal transport package (Flamary et al., 2021) to compute.

**Results**  Figure 3 shows that $RW_2$ Sinkhorn algorithm significantly outperforms the regular Sinkhorn regarding running time. As the length of the translation increases, $RW_2$ Sinkhorn enjoys higher numerical stability in high dimensional data. We also test the performance of the $RW_2$ Sinkhorn algorithm on other different distributions; further results are provided in Appendix B.

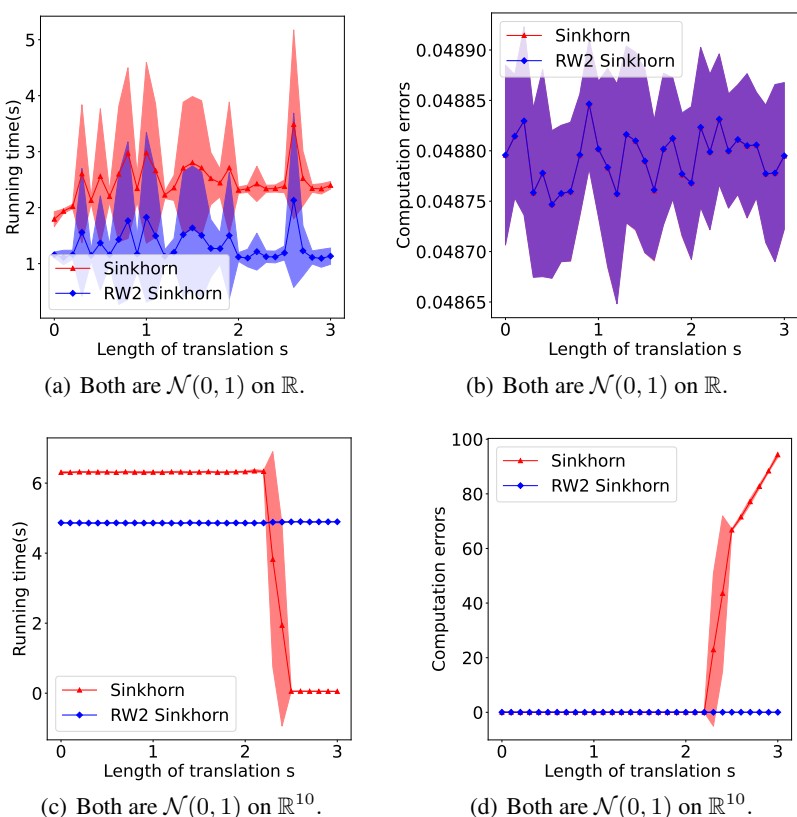

(a) Both are $\mathcal{N}(0,1)$ on $\mathbb{R}$.  (b) Both are $\mathcal{N}(0,1)$ on $\mathbb{R}$.

(c) Both are $\mathcal{N}(0,1)$ on $\mathbb{R}^{10}$.  (d) Both are $\mathcal{N}(0,1)$ on $\mathbb{R}^{10}$.

Figure 3: Comparison of the $RW_2$ Sinkhorn algorithm and the classic Sinkhorn in running time and computational error. When the translation is small, the Sinkhorn algorithm with $RW_2$ technique performs better than the original Sinkhorn algorithm in terms of running time, while keeping almost the same error. As the translation increases, the Sinkhorn algorithm with $RW_2$ technique still enjoys high numerical stability, whereas error explodes in the regular Sinkhorn algorithm.

## 5.2 THUNDERSTORM PATTERN DETECTION

We apply $RW_2$ and general $RW_p$ distances on the real-world thunderstorm dataset, to show that $RW_2$ and general $RW_p$ can be used for identifying similar weather patterns and focus more on shape similarity compared with $W_2$ distance. Our data are radar images from MULTI-RADAR/MULTI-SENSOR SYSTEM (MRMS) (Zhang et al., 2016) in a $300 \times 300\ km^2$ rectangular area centered at the Dallas Fort Worth International Airport (DFW), where each pixel represents a $3 \times 3\ km^2$ area. The data is assimilated every 10 minutes tracking time from 2016 to 2022, with 205,848 images in total. Vertically Integrated Liquid Density (VIL density) and reflectivity are two common measurements for assessing thunderstorm intensity, with threshold values of $3kg \cdot m^{-3}$ and $35dBZ$, respectively (Matthews & Delaura, 2010). We use reflectivity as the main thunderstorm intensity.

We analyze two types of thunderstorm events: snapshots and sequences. Due to page limitation, only the results for thunderstorm snapshots are presented, and the results of thunderstorm sequences are provided in Appendix D.2.

**Settings**  We compute $RW_p$ distances, $p = \{1,2\}$, by the $RW_p$ algorithm and $RW_2$ Sinkhorn algorithm, identifying the top five most similar thunderstorms to a reference event, and compare them with $W_2$. The $RW_2$ Sinkhorn is set with $\lambda = 0.1$ and $\epsilon = 0.01$, and **ot.emd2()** from python optimal transport package (Flamary et al., 2021) is used as the couplings solver for The $RW_p$ algorithm.

Additionally, the resolution of the intermediate radar images for retrieving has been downsampled to $20 \times 20$ pixels to increase computational speed.

**Snapshot results**  Figure 4 demonstrates that, for the same reference thunderstorm snapshot, the top five most similar events identified by $RW_p$ emphasize shape similarity more than those identified by $W_2$. The pattern retrieved by $RW_1$ exhibits more outliers (points significantly distant from the main region) compared to those retrieved by $RW_2$. $RW_2$ offers a balanced consideration of both shape and distance.

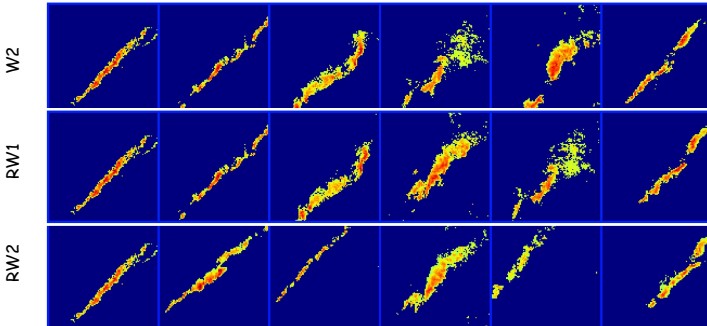

Figure 4: Thunderstorm snapshot comparison using $W_2$ and $RW_p$, $(p = \{1, 2\})$. The leftmost images in the first column are the same reference thunderstorm events. The rest images show the top five most similar thunderstorm snapshots identified by $W_2$ and $RW_p$, sorted in order of distances. The pattern retrieved by $RW_1$ exhibits more outliers (points significantly distant from the main region) compared to those retrieved by $RW_2$, (for example, the fifth picture of $RW_1$ row). $RW_2$ offers a balanced consideration of both shape and distance.

## 6  CONCLUSIONS

In this paper, we introduce a new family of distances, relative-translation invariant Wasserstein ($RW_p$) distances, for measuring the pattern similarity between two probability distributions (and their data supports). Generalizing from the classical optimal transport model, we show that the proposed $RW_p$ distances are real distance metrics defined on the quotient set $\mathcal{P}_p(\mathbb{R}^n)/\sim$ and invariant to the translations. When $p = 2$, this distance enjoys more useful properties, including decomposability of the ROT model and translation-invariance of coupling solutions and $RW_2$. Based on these properties, we show a distribution shift, measured by $W_2$ distance, which can be explained from the perspective of bias-variance. In addition, we propose our algorithm for general $RW_p$ distances and $RW_2$ Sinkhorn algorithm, for efficiently calculating $RW_2$ distance, coupling solutions, as well as $W_2$ distance. We provide the analysis of numerical stability and time complexity for the proposed algorithms. Finally, we validate the $RW_p$ distance and the algorithm performance with illustrative and real-world experiments. The experimental results report that our proposed algorithm significantly improves the computational efficiency of Sinkhorn in practical applications with large translations, and the $RW_2$ distance is robust to distribution translations.

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

# Appendix

## A PROOFS OF THEOREMS

### A.1 PROOF OF THEOREM 1

*Proof of Theorem 1.* It is clear to verify that when $\|s\|_p \geq 2\max_{ij}\|x_i - y_j\|_p$, for any $i, j$, $(1 \leq i \leq m_1, 1 \leq j \leq m_2)$, it follows that $\|x_i + s - y_j\|_p \geq \|s\|_p - \|x_i - y_j\|_p \geq 2\max_{ij}\|x_i - y_j\|_p - \|x_i - y_j\|_p \geq \|x_i - y_j\|_p$. In other words, when $\|s\|_p \geq 2\max_{ij}\|x_i - y_j\|_p$, the relative distance between each pair of support $x_i$ and $y_j$ are always greater than or equal to the non-translated distance, which implies the total transport cost for the translated case will also greater than or equal to the cost for the non-translated distance. Since we are trying to find the minimal value, we can only focus on the compact set $\{s \in \mathbb{R} | \|s\|_p \leq 2\max_{ij}\|x_i - y_j\|_p\}$. $\qquad\square$

### A.2 THEOREM 2

*Proof of Theorem 2.* With the previous notations, firstly, we will show that the translation relation $\sim$ is an equivalence relation on set $\mathcal{P}_p(\mathbb{R})$.

Equivalence relation requires reflexivity, symmetry, and transitivity, and the following observations show translation relation is indeed an equivalence relation.

- Reflexivity, $(x \sim x)$.

  For any distribution $\mu \in \mathcal{P}_p(\mathbb{R}^n)$, it can translate to itself with zero vector.

- Symmetry, $(x \sim y \implies y \sim x)$.

  For any distribution $\mu, \nu \in \mathcal{P}_p(\mathbb{R}^n)$, if $\mu$ can be translated to $\nu$, then $\nu$ can also be translated to $\mu$.

- Transitivity, $(x \sim y \text{ and } y \sim z \implies x \sim z)$.

  For any distribution $\mu, \nu, \eta \in \mathcal{P}_p(\mathbb{R}^n)$, if $\mu$ can be translated to $\nu$, and $\nu$ can be translated to $\eta$, then $\mu$ can also be translated to $\eta$.

Based on the property of equivalence relation, it is clear that set $\mathcal{P}_p(\mathbb{R}^n)/\sim$ is well-defined. Let $[\mu]$ be an element in set $M/\sim$, where $\mu$ is a representative of $[\mu]$, i.e. $[\mu]$ is the set of distributions that can be mutually translated from $\mu$. Noticing that $W_p(\cdot, \cdot)$ is a real distance metric on $\mathcal{P}_p(\mathbb{R}^n)$ (Villani & Society, 2003), it implies that $W_p(\cdot, \cdot)$ satisfied with identity, positivity, symmetry, and the triangle inequality. Based on $W_p(\cdot, \cdot)$, we show $RW_p(\cdot, \cdot)$ satisfies with identity, positivity, symmetry, and triangle inequality w.r.t. elements in $\mathcal{P}_p(\mathbb{R}^n)/\sim$.

For any $\mu, \nu, \eta \in \mathcal{P}_p(\mathbb{R}^n)/\sim$,

- Identity,

$$RW_p([\mu], [\mu]) = \min_{\mu \in [\mu], \mu \in [\mu]}[W_p(\mu, \mu)] = 0.$$

- Positivity,

$$RW_p([\mu], [\nu]) = \min_{\mu \in [\mu], \nu \in [\nu]}[W_p(\mu, \nu)] \geq 0.$$

- Symmetry,

$$RW_p(\mu, \nu) = \min_{\mu \in [\mu], \nu \in [\nu]}[W_p(\mu, \nu)] = \min_{\nu \in [\nu], \mu \in [\mu]}[W_p(\nu, \mu)] = RW_p(\nu, \mu).$$

- Triangle inequality,

$$
\begin{aligned}
RW_p(\mu, \nu) &= \min_{\mu \in [\mu], \nu \in [\nu]} [W_p(\mu, \nu)] \\
&\leq \min_{\eta, \eta' \in [\eta]} \min_{\mu \in [\mu], \nu \in [\nu]} [W_p(\mu, \eta) + W_p(\eta, \eta') + W_p(\eta', \nu)] \\
&= \min_{\mu \in [\mu], \nu \in [\nu], \eta, \eta' \in [\eta]} [W_p(\mu, \eta) + 0 + W_p(\eta', \nu)] \\
&= \min_{\mu \in [\mu], \eta \in [\eta]} [W_p(\mu, \eta)] + \min_{\nu \in [\nu], \eta' \in [\eta]} [W_p(\eta', \nu)] \\
&= \min_{\mu \in [\mu], \eta \in [\eta]} [W_p(\mu, \eta)] + \min_{\nu \in [\nu], \eta \in [\eta]} [W_p(\eta, \nu)] \\
&= RW_p(\mu, \eta) + RW_p(\eta, \nu).
\end{aligned}
$$

$\square$

## A.3 Theorem 3

*Proof of Theorem 3.* Given $P$, because $E_p(s, P)$ is a convex function w.r.t. variable $s$, the minimum $s$ must satisfy with

$$
\frac{\partial E_p(s, P)}{\partial s} = \sum_{i=1}^{m_1} \sum_{j=1}^{m_2} p P_{ij} \operatorname{sign}(x_i + s - y_j) \|x_i + s - y_j\|_p^{p-1} = 0. \tag{8}
$$

For the outer function $F(P) = \min_{s \in \mathbb{R}^n} E_p(s, P)$, we can remove $\min_{s \in \mathbb{R}^n}$ by using the equivalent constraint $\frac{\partial E_p(s, P)}{\partial s} = 0$, i.e.,

$$
F(P) = \min_{s \in \mathbb{R}^n} E_p(s, P) = E_p(s_P, P)\big|_{\frac{\partial E_p(s_P, P)}{\partial s} = 0}. \tag{9}
$$

Therefore,

$$
\begin{aligned}
\frac{\partial F(P)}{\partial P_{ij}} &= \frac{\partial E_p(s_P, P)}{\partial P_{ij}} \\
&= \frac{\partial E_p}{\partial s_P} \frac{\partial s_P}{\partial P_{ij}} + \frac{\partial E_p}{\partial P_{ij}} \\
&= 0 \times \frac{\partial s_P}{\partial P_{ij}} + \|x_i + s - y_j\|_p^p \\
&= \|x_i + s - y_j\|_p^p.
\end{aligned}
$$

$$(10)$$

$\square$

## A.4 Proof of Theorem 4

*Proof of Theorem 4.* With the previous notations, firstly, we show the two-stage optimization problem, $\min_{s \in \mathbb{R}^n} \min_{P \in \Pi(\mu, \nu)} E_2(s, P)$, can be decomposed into two independent one-stage optimization problems, $\min_{P \in \Pi(\mu, \nu)} H(P)$ and $\min_{s \in \mathbb{R}^n} V(s)$.

For the objective function $E_2(s, P)$, we expand it w.r.t. $s$,

$$E_2(s, P)$$
$$= \sum_{i=1}^{m_1} \sum_{j=1}^{m_2} P_{ij} \|x_i + s - y_j\|_2^2$$
$$= \sum_{i=1}^{m_1} \sum_{j=1}^{m_2} P_{ij} \left( \|x_i - y_j\|_2^2 + \|s\|_2^2 + 2s \cdot (x_i - y_j) \right) \tag{11}$$
$$= \sum_{i=1}^{m_1} \sum_{j=1}^{m_2} P_{ij} \|x_i - y_j\|_2^2 + \sum_{i=1}^{m_1} \sum_{j=1}^{m_2} P_{ij} \|s\|_2^2 + 2 \sum_{i=1}^{m_1} \sum_{j=1}^{m_2} P_{ij} s \cdot (x_i - y_j).$$

We can rewrite the second and the third terms in Equation (11) under the condition $P \in \Pi(\mu, \nu)$, which implies that,

$$\sum_{i=1}^{m_1} \sum_{j=1}^{m_2} P_{ij} = 1, \sum_{j=1}^{m_2} P_{ij} = a_i, \sum_{i=1}^{m_1} P_{ij} = b_j, 1 \le i \le m_1, 1 \le j \le m_2.$$

For the second term, it follows that

$$\sum_{i=1}^{m_1} \sum_{j=1}^{m_2} P_{ij} \|s\|_2^2 = \|s\|_2^2 \cdot (\sum_{i=1}^{m_1} \sum_{j=1}^{m_2} P_{ij}) = \|s\|_2^2 \cdot 1 = \|s\|_2^2.$$

For the third term, it follows that

$$2 \sum_{i=1}^{m_1} \sum_{j=1}^{m_2} P_{ij} s \cdot (x_i - y_j)$$
$$= 2s \cdot \sum_{i=1}^{m_1} \sum_{j=1}^{m_2} P_{ij} (x_i - y_j)$$
$$= 2s \cdot (\sum_{i=1}^{m_1} \sum_{j=1}^{m_2} x_i \cdot P_{ij} - \sum_{i=1}^{m_1} \sum_{j=1}^{m_2} y_j \cdot P_{ij})$$
$$= 2s \cdot (\sum_{i=1}^{m_1} x_i \cdot (\sum_{j=1}^{m_2} P_{ij}) - \sum_{j=1}^{m_2} y_j \cdot (\sum_{i=1}^{m_1} P_{ij}))$$
$$= 2s \cdot (\sum_{i=1}^{m_1} x_i \cdot a_i - \sum_{j=1}^{m_2} y_j \cdot b_j)$$
$$= 2s \cdot (\bar{\mu} - \bar{\nu}).$$

Thus, we have the following transformation,

$$\min_{s \in \mathbb{R}^n} \min_{P \in \Pi(\mu, \nu)} E_2(P, s)$$
$$= \min_{s \in \mathbb{R}^n} \min_{P \in \Pi(\mu, \nu)} (\sum_{i=1}^{m_1} \sum_{j=1}^{m_2} \|x_i - y_j\|_2^2 P_{ij} + \sum_{i=1}^{m_1} \sum_{j=1}^{m_2} \|s\|_2^2 P_{ij} + 2 \sum_{i=1}^{m_1} \sum_{j=1}^{m_2} s \cdot (x_i - y_j) P_{ij})$$
$$= \min_{s \in \mathbb{R}^n} \min_{P \in \Pi(\mu, \nu)} \sum_{i=1}^{m_1} \sum_{j=1}^{m_2} \|x_i - y_j\|_2^2 P_{ij} + \min_{s \in \mathbb{R}^n} \min_{P \in \Pi(\mu, \nu)} (\sum_{i=1}^{m_1} \sum_{j=1}^{m_2} \|s\|_2^2 P_{ij} + 2 \sum_{i=1}^{m_1} \sum_{j=1}^{m_2} s \cdot (x_i - y_j) P_{ij})$$
$$= \min_{s \in \mathbb{R}^n} \min_{P \in \Pi(\mu, \nu)} \sum_{i=1}^{m_1} \sum_{j=1}^{m_2} \|x_i - y_j\|_2^2 P_{ij} + \min_{s \in \mathbb{R}^n} (\|s\|_2^2 + 2s \cdot (\bar{\mu} - \bar{\nu}))$$
$$= \min_{P \in \Pi(\mu, \nu)} \sum_{i=1}^{m_1} \sum_{j=1}^{m_2} \|x_i - y_j\|_2^2 P_{ij} + \min_{s \in \mathbb{R}^n} (\|s\|_2^2 + 2s \cdot (\bar{\mu} - \bar{\nu}))$$
$$= \min_{P \in \Pi(\mu, \nu)} H(P) + \min_{s \in \mathbb{R}^n} V(s)$$

Since $V(s)$ is a quadratic function of variable $s$, it is easy to follow that the minimum is achieved when $s = \bar{\nu} - \bar{\mu}$.

$\square$

## B    COMPLEXITY ANALYSIS FOR $RW_2$ ALGORITHM UNDER SUB-GAUSSIAN DISTRIBUTIONS

This section is organized as follows. In section B.1, we state and prove the theorem regarding the time complexity of $RW_2$ Algorithm. We leave the definitions and theorems used in the proof to section B.2.

### B.1    THEORETICAL RESULTS OF TIME COMPLEXITY

*Proof of Theorem 5.* For $i = 1, 2, \ldots, m_1$, $X_i - \bar{\mu}$ is a sub-Gaussian random vector. Using Theorem 7 and taking a union bound over all the random vectors, we have for all $X_i$ with probability at least $1 - \delta/4$, the following inequality holds

$$\|X_i - \bar{\mu}\|_2 \leq c\big(\sqrt{n} + \sqrt{\log(4m_1/\delta)}\big) \cdot \|\mu - \bar{\mu}\|_{\psi_2}. \tag{12}$$

Similarly, we have for all $Y_j$, with probability at least $1 - \delta$, the following inequality holds

$$\|Y_j - \bar{\nu}\|_2 \leq c\big(\sqrt{n} + \sqrt{\log(4m_2/\delta)}\big) \cdot \|\nu - \bar{\nu}\|_{\psi_2}. \tag{13}$$

Using Theorem 6, $\sum_{i=1}^m (X_i - \bar{\mu})$ is a sub-Gaussian random vector, with $\|\sum_{i=1}^{m_1}(X_i - \bar{\mu})\|_{\psi_2} \leq \sqrt{L \sum_{i=1}^{m_1} \|X_i - \bar{\mu}\|_{\psi_2}^2}$ Then using Theorem 7, with probability at least $1 - \delta/4$, we have

$$\Big\| \sum_{i=1}^{m_1} X_i - m_1 \bar{\mu} \Big\|_2 \leq c\big(\sqrt{n} + \sqrt{\log(1/\delta)}\big) \cdot \Big\| \sum_{i=1}^{m_1}(X_i - \bar{\mu}) \Big\|_{\psi_2}$$

$$= c\big(\sqrt{n} + \sqrt{\log(1/\delta)}\big) \cdot \sqrt{L \sum_{i=1}^{m_1} \|X_i - \bar{\mu}\|_{\psi_2}^2}$$

$$= c'\big(\sqrt{n} + \sqrt{\log(1/\delta)}\big) \cdot \sqrt{m_1} \|\mu - \bar{\mu}\|_{\psi_2}, \tag{14}$$

where $c'$ is an absolute constant. Similarly, with probability at least $1 - \delta$, we have

$$\Big\| \sum_{j=1}^{m_2} Y_j - m_2 \bar{\nu} \Big\|_2 \leq c'\big(\sqrt{n} + \sqrt{\log(1/\delta)}\big) \cdot \sqrt{m_2} \|\nu - \bar{\nu}\|_{\psi_2}, \tag{15}$$

where $c'$ is an absolute constant. In the following proof, we consider the union bound of all the high-probability events above, such that (12), (13), (14) and (15) hold. It occurs with probability at least $1 - \delta$.

First, for $\max_{i,j} \|X_i - Y_j\|_2$, we have

$$\max_{i,j} \|X_i - Y_j\|_2 \geq \max_{i,j} \|\bar{\mu} - \bar{\nu}\|_2 - \|X_i - \bar{\mu}\|_2 - \|Y_j - \bar{\nu}\|_2$$

$$\geq l - \left[ c\big(\sqrt{n} + \sqrt{\log(4m_1/\delta)}\big) \cdot \|\mu - \bar{\mu}\|_{\psi_2} \right]$$

$$- \left[ c\big(\sqrt{n} + \sqrt{\log(4m_2/\delta)}\big) \cdot \|\nu - \bar{\nu}\|_{\psi_2} \right]$$

$$= l - 2c\sqrt{n} - c\sqrt{\log(4m_1/\delta)}\big) \cdot \|\mu - \bar{\mu}\|_{\psi_2}$$

$$- c\sqrt{\log(4m_2/\delta)}\big) \cdot \|\nu - \bar{\nu}\|_{\psi_2},$$

where the first inequality holds due to the triangle inequality. The second inequality holds due to (12) and (13).

For $\max_{i,j} \|X_i - Y_j - \bar{X}_i + \bar{Y}_j\|_2$, we have

$$\max_{i,j} \|X_i - Y_j - \bar{X}_i + \bar{Y}_j\|_2 \leq \max_{i,j} \|X_i - \mu\|_2 + \|Y_j - \nu\|_2$$
$$+ \|\bar{X}_i - \mu\|_2 + \|\bar{Y}_j - \nu\|_2$$
$$\leq \left[ c\left(\sqrt{n} + \sqrt{\log(4m_1/\delta)}\right) \cdot \|\mu - \bar{\mu}\|_{\psi_2} \right] + \left[ c\left(\sqrt{n} + \sqrt{\log(4m_2/\delta)}\right) \cdot \|\nu - \bar{\nu}\|_{\psi_2} \right]$$
$$+ \left[ c' \frac{\sqrt{n} + \sqrt{\log(1/\delta)}}{\sqrt{m_1}} \|\mu - \bar{\mu}\|_{\psi_2} \right] + \left[ c' \frac{\sqrt{d} + \sqrt{\log(1/\delta)}}{\sqrt{m_2}} \|\nu - \bar{\nu}\|_{\psi_2} \right]$$
$$\leq L\sqrt{n}\left[ 1 + \frac{\|\mu - \bar{\mu}\|_{\psi_2}}{\sqrt{m_1}} + \frac{\|\nu - \bar{\nu}\|_{\psi_2}}{\sqrt{m_2}} \right]$$
$$+ L\left[ \sqrt{\log(4m_1/\delta)} \cdot \|\mu - \bar{\mu}\|_{\psi_2} + \sqrt{\log(4m_2/\delta)} \cdot \|\nu - \bar{\nu}\|_{\psi_2} \right],$$

where the first inequality holds due to (12), (13), (14) and (15). Therefore, we have the following conclusion: As long as

$$l \geq L\sqrt{n}\left[ 1 + \|\mu - \bar{\mu}\|_{\psi_2} + \|\nu - \bar{\nu}\|_{\psi_2} \right]$$
$$+ L\left[ \sqrt{\log(4m_1/\delta)} \cdot \|\mu - \bar{\mu}\|_{\psi_2} + \sqrt{\log(4m_2/\delta)} \cdot \|\nu - \bar{\nu}\|_{\psi_2} \right],$$

where $L$ is an absolute constant, we can conclude that

$$\max_{i,j} \|X_i - Y_j - \bar{X}_i + \bar{Y}_j\|_2 \leq \max_{i,j} \|X_i - Y_j\|_2.$$

This completes the proof of Theorem 5. $\qquad\square$

## B.2  HIGH DIMENSIONAL PROBABILITY BASICS

In this section, we introduce some basic knowledge we have used in the proof of Theorem 5. The results mainly come from Vershynin (2018).
We first introduce a broad and widely used distribution class.

**Definition 5** (Sub-Gaussian). *A random variable $X$ that satisfies one of the following equivalent properties is called a subgaussian random variable.*

(a) *There exists $K_1 > 0$ such that the tails of $X$ satisfy*

$$\mathbb{P}\{|X| \geq t\} \leq 2\exp(-t^2/K_1^2) \text{ for all } t \geq 0.$$

(b) *There exists $K_2 > 0$ such that the moments of $X$ satisfy*

$$\|X\|_{L^p} = (\mathbb{E}|X|^p)^{1/p} \leq K_2\sqrt{p} \text{ for all } p \geq 1.$$

(c) *There exists $K_3 > 0$ such that the moment-generating function (MGF) of $X^2$ satisfies*

$$\mathbb{E}\exp(\lambda^2 X^2) \leq \exp(K_3^2\lambda^2) \text{ for all } \lambda \text{ such that } |\lambda| \leq \frac{1}{K_3}.$$

(d) *There exists $K_4 > 0$ such that the MGF of $X^2$ is bounded at some point, namely,*

$$\mathbb{E}\exp(X^2/K_4^2) \leq 2.$$

(e) *Moreover, if $\mathbb{E}X = 0$, the following property is also equivalent. There exists $K_5 > 0$ such that the MGF of $X$ satisfies*

$$\mathbb{E}\exp(\lambda X) \leq \exp(K_5^2\lambda^2) \text{ for all } \lambda \in \mathbb{R}.$$

The parameters $K_i > 0$ appearing in these properties differ from each other by at most an absolute constant factor.

The sub-gaussian norm of $X$, denoted $\|X\|_{\psi_2}$, is defined to be

$$\|X\|_{\psi_2} = \inf\{t > 0 : \mathbb{E}\exp(X^2/t^2) \leq 2\}.$$

**Definition 6.** *A random vector $X \in \mathbb{R}^d$ is sub-Gaussian if for any vector $\mathbf{u} \in \mathbb{R}^d$ the inner product $\langle X, \mathbf{u} \rangle$ is a sub-Gaussian random variable. And the corresponding $\psi_2$ norm of $X$ is defined as*

$$\|X\|_{\psi_2} = \sup_{\|\mathbf{u}\|_2 = 1} \|\langle X, \mathbf{u} \rangle\|_{\psi_2}.$$

**Theorem 6.** *Let $X_1, \ldots, X_N \in \mathbb{R}^d$ be independent, mean zero, sub-Gaussian random vectors. Then $\sum_{i=1}^N X_i$ is also a sub-Gaussian random vector, and*

$$\Big\| \sum_{i=1}^N X_i \Big\|_{\psi_2}^2 \leq L \sum_{i=1}^N \|X_i\|_{\psi_2}^2.$$

*where $L$ is an absolute constant.*

*Proof of Theorem 6.* For any vector $\mathbf{u} \in \mathbb{R}$, $\|\mathbf{u}\|_2 = 1$, consider $\langle \sum_{i=1}^N X_i, \mathbf{u} \rangle$. Using independence, we have for all $\lambda$,

$$\mathbb{E}\exp\Big(\lambda \sum_{i=1}^N \langle X_i, \mathbf{u} \rangle\Big) = \prod_{i=1}^N \mathbb{E}\exp\big(\lambda \langle X_i, \mathbf{u} \rangle\big)$$

$$\leq \prod_{i=1}^N \exp\big(L\|\langle X_i, \mathbf{u} \rangle\|_{\psi_2}^2 \lambda^2\big)$$

$$= \exp\Big(L\lambda^2 \sum_{i=1}^N \|\langle X_i, \mathbf{u} \rangle\|_{\psi_2}^2\Big),$$

where $L$ is an absolute constant and the first inequality holds due to property (e) of the sub-Gaussian variables. Taking supreme over $\mathbf{u}$, we prove that $\sum_{i=1}^N X_i$ is also a sub-Gaussian random vector. Moreover,

$$\Big\| \sum_{i=1}^N X_i \Big\|_{\psi_2}^2 \leq L \sum_{i=1}^N \|X_i\|_{\psi_2}^2.$$

where L is an absolute constant. $\qquad \square$

**Theorem 7.** *Let $X \in \mathbb{R}^d$ be a sub-Gaussian random vector. Then with probability at least $1 - \delta$,*

$$\|X\|_2 \leq c\big(\sqrt{d} + \sqrt{\log(1/\delta)}\big) \cdot \|X\|_{\psi_2}.$$

*Proof.* Let $B_d$ be the $d$-dimensional unit ball, $N$ be a $1/2$-covering of $B_d$ in 2-norm with covering number $= N(B_d, \|\cdot\|_2, 1/2)$. Therefore,

$$\forall \mathbf{x} \in B_d, \exists \mathbf{z} \in N, \text{ s.t. } \|\mathbf{x} - \mathbf{z}\| \leq 1/2.$$

Using Lemma 1, we have

$$N \leq 5^d. \tag{16}$$

Using the fact $\|\mathbf{x}\|_2 = \max_{\|\mathbf{y}\|_2 \leq 1} \langle \mathbf{x}, \mathbf{y} \rangle$, we have

$$\|X\|_2 = \max_{\mathbf{x} \in B_d} \langle \mathbf{x}, X \rangle$$

$$\leq \max_{\mathbf{z} \in N} \langle \mathbf{z}, X \rangle + \max_{\mathbf{y} \in (1/2)B_d} \langle \mathbf{y}, X \rangle$$

$$= \max_{\mathbf{z} \in N} \langle \mathbf{z}, X \rangle + \frac{1}{2}\max_{\mathbf{y} \in B_d} \langle \mathbf{y}, X \rangle.$$

Therefore, we have

$$\|X\|_2 \leq 2 \max_{\mathbf{z} \in N} \langle \mathbf{z}, X \rangle. \tag{17}$$

Then we can provide a high probability upper bound for the Euclidean norm of the random vector $X$ by considering the probability $\mathbb{P}(\|X\|_2 \geq t)$.

$$\mathbb{P}(\|X\|_2 \geq t) \leq \mathbb{P}\Big( \max_{\mathbf{z} \in N} \langle \mathbf{z}, X \rangle \geq \frac{t}{2} \Big)$$

$$\leq \mathbb{P}\Big( \exists \mathbf{z} \in N, \langle \mathbf{z}, X \rangle \geq \frac{t}{2} \Big)$$

$$\leq \sum_{\mathbf{z} \in N} \mathbb{P}\Big( \langle \mathbf{z}, X \rangle \geq \frac{t}{2} \Big)$$

$$\leq N \exp\Big( - c \frac{t^2}{\|X\|_{\psi_2}^2} \Big)$$

$$\leq 5^d \exp\Big( - \frac{ct^2}{\|X\|_{\psi_2}^2} \Big),$$

where $c$ is an absolute constant. Here the first inequality holds due to (17). The second inequality holds due to $\{\max_{\mathbf{z} \in N} \langle \mathbf{z}, X \rangle \geq t/2\} \subseteq \{\exists \mathbf{z} \in N, \langle \mathbf{z}, X \rangle \geq t/2\}$. The third inequality holds due to the union bound. The fourth inequality holds due to the definition of the sub-Gaussian vector and the property (a) of a sub-Gaussian variable. The last inequality holds due to (16).

Finally, let $t = \sqrt{[d \log 5 + \log(1/\delta)]/c} \cdot \|X\|_{\psi_2}$. We have with probability at least $1 - \delta$,

$$\|X\|_2 \geq t.$$

Finally, using $\sqrt{a + b} \leq \sqrt{a} + \sqrt{b}$, we complete the proof of Theorem 7. $\square$

**Definition 7** ($\epsilon$-covering). *Let $(V, \|\cdot\|)$ be a normed space, and $\Theta \subset V$. $V_1, \ldots, V_N$ is an $\epsilon$-covering of $\Theta$ if $\Theta \subseteq \cup_{i=1}^N V_i$, or equivalently, $\forall \theta \in \Theta, \exists i$ such that $\|\theta - V_i\| \leq \epsilon$.*

**Definition 8** (Covering number). *The covering number is defined by*

$$N(\Theta, \|\cdot\|, \epsilon) := \min\{n : \exists \epsilon\text{-covering over } \Theta \text{ of size } n\}.$$

**Lemma 1.** *Let $B_d$ be the $d$-dimensional Euclidean unit ball. Consider $N(B_d, \|\cdot\|_2, \epsilon)$. When $\epsilon \geq 1$, $N(B_d, \|\cdot\|_2, \epsilon) = 1$. When $\epsilon < 1$, we have*

$$\Big( \frac{1}{\epsilon} \Big)^d \leq N(B_d, \|\cdot\|_2, \epsilon) \leq \Big( 1 + \frac{2}{\epsilon} \Big)^d.$$

## C  COUNTEREXAMPLES

In this section, we provide several counterexamples to show the outer function $\min_{P \in \Pi(\mu,\nu)} E_p(s, P)$ in the original ROT problem is strictly non-convex w.r.t. the variable $s$ in the high dimensional case. Next, we provide one counterexample to show function $\min_{s \in \mathbb{R}^n} E_p(s, P)$ is non-convex w.r.t. variable $P$. Finally, we show that the optimal translation is not always the same as the difference between the means of two distributions when $p \neq 2$.

### C.1  FUNCTION $\min_{P \in \Pi(\mu,\nu)} E_p(s, P)$

Assume the underlying space is in two-dimensional space and source and target distribution $\mu$ and $\nu$ are formed by $\{x_i = (\cos \frac{2i\pi}{3}, \sin \frac{2i\pi}{3}), i = 1, 2, 3\}$ and $\{y_j = (-\cos \frac{2j\pi}{3}, -\sin \frac{2j\pi}{3}), j = 1, 2, 3\}$ with equal masses, respectively, which is shown in the following figure 5 (a).

First, we will demonstrate that the outer function $\min_{P \in \Pi(\mu,\nu)} E_p(s, P)$ in the original ROT problem is not convex w.r.t. the variable $s$ under the given source and target distributions.

When $p = 1$, by enumerating the values of $s$ over the 100x100 grid in region $[-1.2, 1.2]^2$, we can plot the contour and function values of $\min_{P \in \Pi(\mu,\nu)} E_1(s, P)$ w.r.t. the variable $s$. These results show the non-convexity of function $\min_{P \in \Pi(\mu,\nu)} E_1(s, P)$, which are illustrated in Figures 5 (b) and (c).

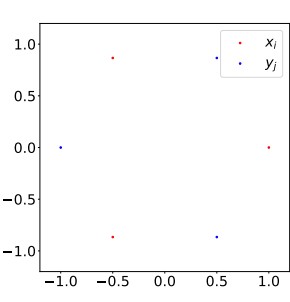 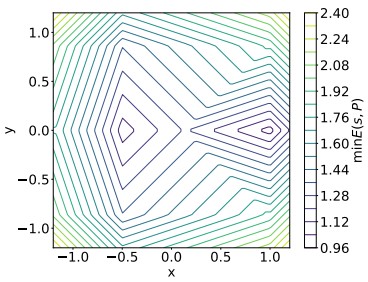 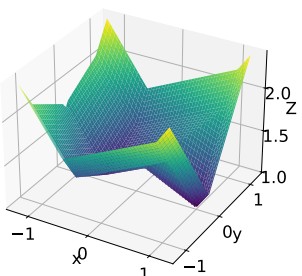

(a) Distribution $\mu$ and $\nu$.

(b) Contour plot of function $\min_{P \in \Pi(\mu,\nu)} E_1(s, P)$ w.r.t. the variable $s$, where the coordinate $(x, y)$ represents the translation $s$.

(c) Value of the inner function w.r.t. the variable $s$, where the coordinate $(x, y)$ represents the translation $s$ and $Z$ is the value of $\min_{P \in \Pi(\mu,\nu)} E_1(s, P)$.

Figure 5: Contourplot and value of function $\min_{P \in \Pi(\mu,\nu)} E_1(s, P)$ w.r.t. the variable $s$, which shows the inner function is non-convex when $p = 1$.

Under the same source and target distributions, we also show the non-convexity of other cases in Figure 6 when $p = \{1.2, 4, 10\}$. The contourplots and values of function $\min_{P \in \Pi(\mu,\nu)} E_p(s, P)$ w.r.t. the variable $s$ show the non-convexity of function $\min_{P \in \Pi(\mu,\nu)} E_p(s, P)$, which are illustrated in Figures 6.

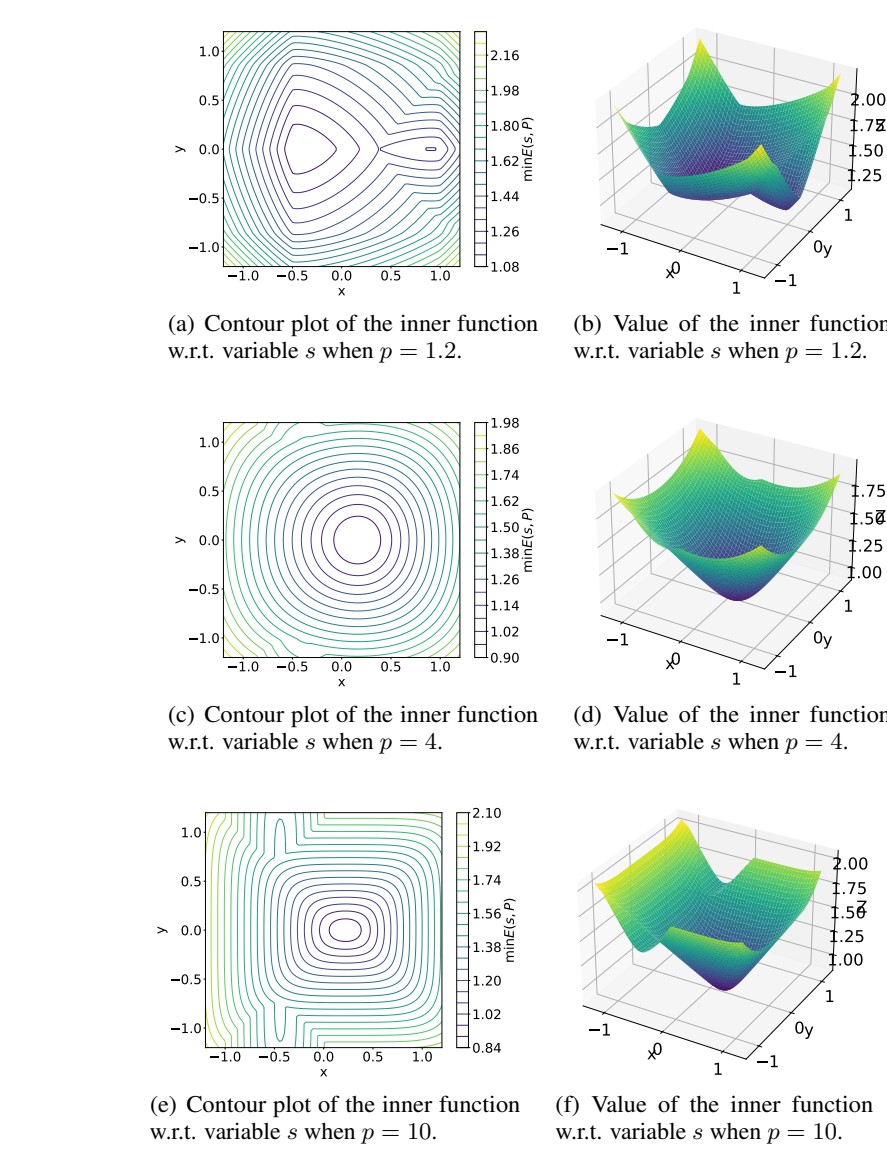

(a) Contour plot of the inner function w.r.t. variable $s$ when $p = 1.2$.

(b) Value of the inner function w.r.t. variable $s$ when $p = 1.2$.

(c) Contour plot of the inner function w.r.t. variable $s$ when $p = 4$.

(d) Value of the inner function w.r.t. variable $s$ when $p = 4$.

(e) Contour plot of the inner function w.r.t. variable $s$ when $p = 10$.

(f) Value of the inner function w.r.t. variable $s$ when $p = 10$.

Figure 6: Contourplot and value of function $\min\limits_{P \in \Pi(\mu,\nu)} E_1(s, P)$ w.r.t. the variable $s$, which shows the inner function is non-convex when $p = \{1.2, 4, 10\}$.

## C.2 FUNCTION $\min\limits_{s \in \mathbb{R}^n} E_p(s, P)$

Next, under the given source and target distributions and assume $p = 1$, we demonstrate that function $F_1(P) = \min\limits_{s \in \mathbb{R}^n} E_1(s, P)$ in the reformulated ROT problem is also not convex w.r.t. the variable $s$.

Let us consider two transport plan, $P_1$ and $P_2$, where $P_1 = \frac{1}{3} \begin{bmatrix} 1 & 0 & 0 \\ 0 & 0 & 1 \\ 0 & 1 & 0 \end{bmatrix}$ and $P_2 = \frac{1}{3} \begin{bmatrix} 0 & 1 & 0 \\ 0 & 0 & 1 \\ 1 & 0 & 0 \end{bmatrix}$.

It is easy to verify that the minimizers of $F_1(P_1) = \min\limits_{s \in \mathbb{R}^n} E_1(s, P_1)$ and $F_1(P_2) = \min\limits_{s \in \mathbb{R}^n} E_1(s, P_2)$ are $s_{P_1} = (1, 0)$ and $s_{P_2} = (-0.5, 0)$, respectively. Consequently, we can compute $F_1(P_1) = 1$ and $F_1(P_2) = \frac{1}{2} + \frac{1\sqrt{3}}{3}$. Notice that $F_1(\frac{1}{2}P_1 + \frac{1}{2}P_2) = 1 + \frac{\sqrt{3}}{6} > \frac{1}{2}(1 + \frac{1}{2} + \frac{1\sqrt{3}}{3}) = \frac{1}{2}F_1(P_1) + \frac{1}{2}F_1(P_2)$, therefore, $F_1(P)$ is not convex w.r.t. variable $P$.

## C.3 THE OPTIMAL RELATIVE TRANSLATION

In the following, we show that the optimal relative translation is not always the same as the difference between the means of two distributions when $p \neq 2$.

Assume the underlying space is in two-dimensional space and source and target distribution $\mu$ and $\nu$ are formed by $\{x_1 = (3, 0), x_2 = (0, 0), x_3 = (0, 3)\}$ and $\{y_1 = (-3, 0), y_2 = (0, 0), y_3 = (0, -3)\}$ with equal masses, respectively.

Consider the case when $p = 1$. Since the mass center (centroid) of distribution $\mu$ and $\nu$ in terms of $L_1$ norm are $\bar{\mu} = (0, 0)$ and $\bar{\nu} = (0, 0)$, if we take their difference as a translation, we can get $W_1(\mu, \nu) = \frac{(3+3+6)}{3} = 4$. However, this translation is not optimal, since when the translation $s_0 = (-3, -3)$, the total transport cost is $W_1(\mu + s_0, \nu) = \frac{(3+3)}{3} = 2 < W_1(\mu, \nu)$. Therefore, the optimal translation might not be the difference between the means of two distributions, when $p \neq 2$.

# D ADDITIONAL EXPERIMENT RESULTS

## D.1 ADDITIONAL EXPERIMENT RESULTS FOR SECTION 5.1 - NUMERICAL VALIDATION

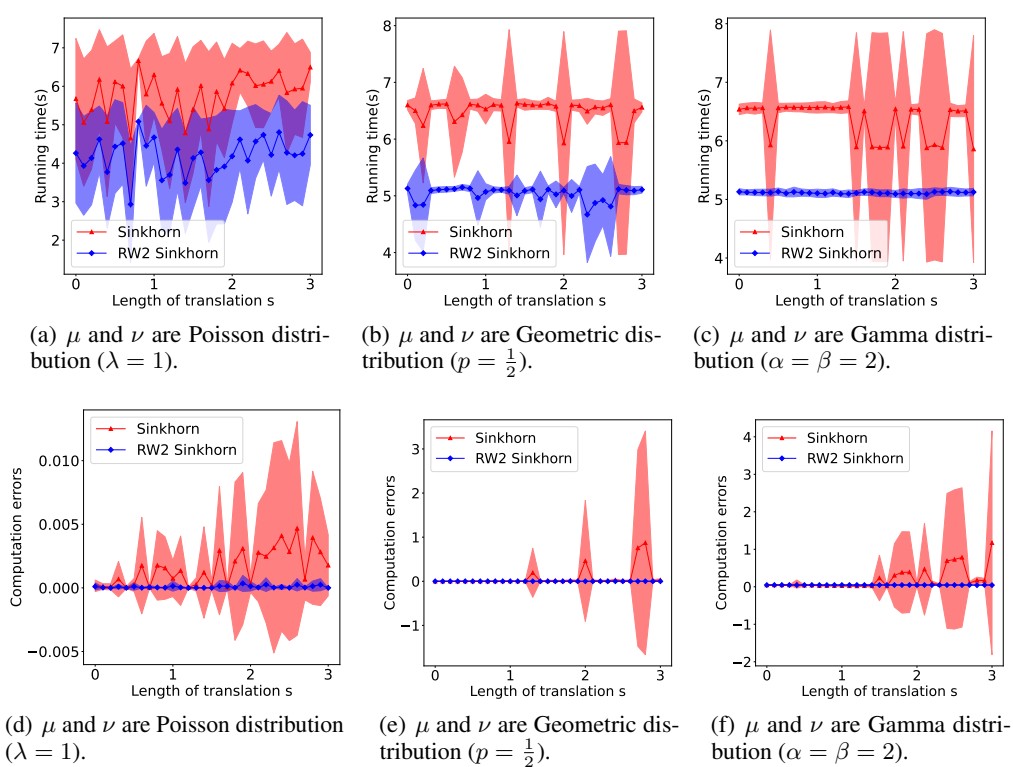

(a) $\mu$ and $\nu$ are Poisson distribution ($\lambda = 1$).

(b) $\mu$ and $\nu$ are Geometric distribution ($p = \frac{1}{2}$).

(c) $\mu$ and $\nu$ are Gamma distribution ($\alpha = \beta = 2$).

(d) $\mu$ and $\nu$ are Poisson distribution ($\lambda = 1$).

(e) $\mu$ and $\nu$ are Geometric distribution ($p = \frac{1}{2}$).

(f) $\mu$ and $\nu$ are Gamma distribution ($\alpha = \beta = 2$).

Figure 7: Additional results from the experiment in Section 5.1. The first column shows the results from a pair of Poisson distributions, the second column shows the results from a pair of Geometric distributions, and the third column shows the results from a pair of Gamma distributions, all of which are defined on $\mathbb{R}$.

## D.2 ADDITIONAL EXPERIMENT RESULTS FOR SECTION 5.2 - SIMILAR THUNDERSTORM PATTERN DETECTION

**Snapshot results** Figure 8 shows the snapshot comparison between $RW_2$ and $W_2$ for other different references.

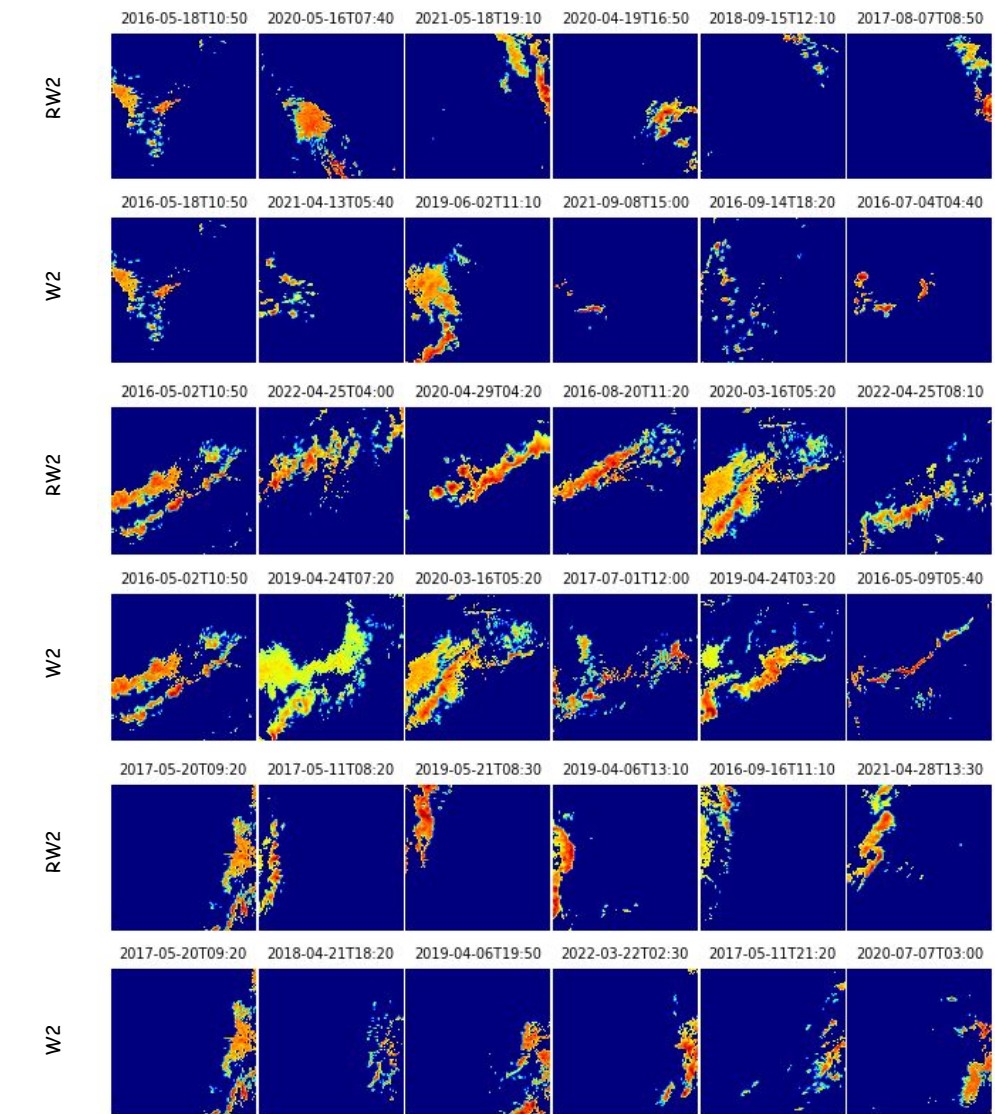

Figure 8: Additional examples of similar thunderstorm snapshot identification using $RW_2$ and $W_2$. The leftmost images in the first column are the reference thunderstorm events, which are 2016-05-18-10:50, 2016-05-02-10:50, and 2017-05-20-09:20. The other images show the top 5 most similar thunderstorm snapshots identified by $RW_2$ and $W_2$, sorted in order of similarity.

**Sequence settings**    Similar to the comparison of individual snapshots, a sequence of thunderstorm events (a series of thunderstorm snapshots) can also be treated as a probability distribution by incorporating time as a third-dimensional axis. Given that temporal information is independent of spatial information, we set the temporal-spatial tradeoff to 1 to balance both information. We present only the results for $W_2$ and $RW_2$ distances since retrieving results.

**Sequence results**    Figure 9 presents the results of identifying similar thunderstorm sequences using $RW_2$ and $W_2$. The first row shows the reference thunderstorm sequence, which lasts for 1 hour. The second through fifth rows display the top four most similar sequences identified by $RW_2$, while the sixth through ninth rows show the top four most similar sequences identified by $W_2$. Once again, it is evident that $RW_2$ prioritizes pattern (shape) similarity, whereas $W_2$ tends to be influenced by location similarity.

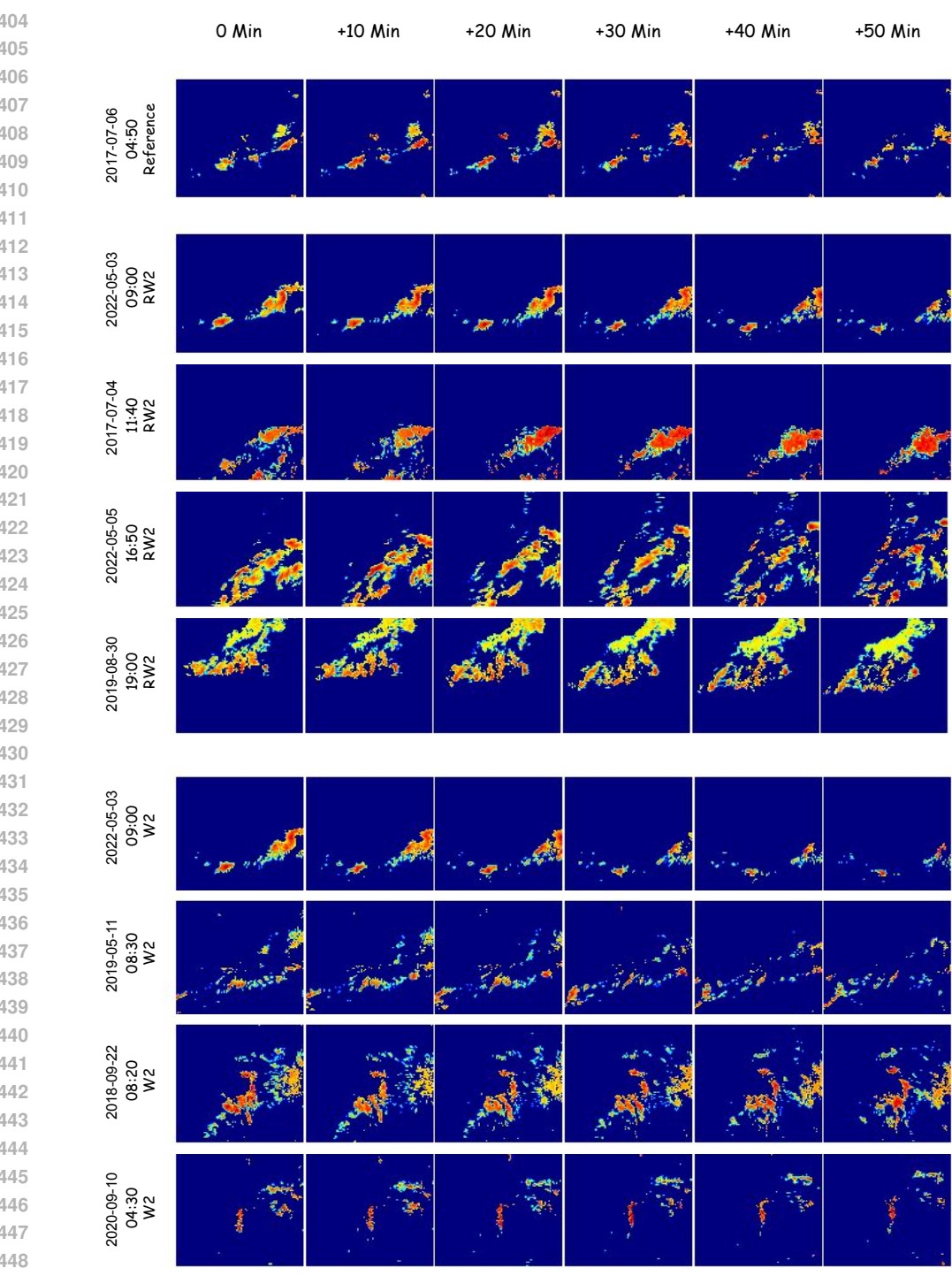

Figure 9: Similar thunderstorm sequence identification using $RW_2$ and $W_2$. The first row is the reference thunderstorm sequence with a 1-hour duration. The second to the fifth rows are the top four most similar thunderstorm sequences identified by $RW_2$. The sixth to the ninth rows are the top four most similar thunderstorm sequences identified by $W_2$. Again we observe that $RW_2$ focuses more on pattern (shape) similarity, and $W_2$ gets distracted by location similarity.

