# OpenReview forum: "Relative-Translation Invariant Wasserstein Distance"
_ICLR.cc/2025/Conference — Submitted to ICLR 2025_

### Official Review · Reviewer_EYZ9 · 2024-10-24

**Soundness:** 2
**Presentation:** 3
**Contribution:** 2
**Rating:** 3
**Confidence:** 4

**Summary:**

This paper proposed a relative-translation invariant Wasserstein distance $RW_p$. Two algorithms were proposed to compute the RW distance: a two-stage algorithm for the general $p$ case, and a variant Sinkhorn algorithm to compute $RW_2$.

**Strengths:**

Classical Wasserstein distance $W_p$ cannot identify similar "shape" patterns in data when their exact locations are irrelevant. Motivated by detecting weather patterns from geographical images and robustness in distributional shift, the paper proposed a new distance metric on the space of probability distributions, called the relative-translation invariant Wasserstein ($RW_p$) distance.

The decomposition of the relative translation optimal transport (ROT) problem into a vertical move along orbit in the quotient space and a horizontal move is interesting. For $p=2$, this decomposition turns out to be an orthogonal "bias-variance" structure of the $W_2$ distance. This effectively generalizes the Bures-Wasserstein distance for Gaussians to translation invariant distributions.

The proposed Algorithm 1 for general $RW_p$ is a coordinate-type algorithm for solving ROT: the vertical component is solved via gradient descent and the horizontal is solved by Sinkhorn or linear programming. When $p=2$, proposed Algorithm 2 is essentially an orbit debiased Sinkhorn algorithm, to avoid numeric instability. Some numerical experiments are performed to demonstrate better numeric stability.

**Weaknesses:**

In Section 4.4, the paper claims that "Consequently, our $RW_2$ method achieves better time complexity compared to $W_2$" in Remark 1 (page 8). I am not convinced why it is so. Theorem 5 only proves that with high probability the largest value of centered pairwise distances between source and target distributions (i.e., worst cost value) gets improved. It is widely unclear why this result alone implies better time or iteration complexity for solving $RW_p$ than $W_p$. In particular, if $p \neq 2$, solving $W_p$ requires only a pass of solving Sinkhorn, while $RW_p$ (with unknown means of source and target) has to solve many Sinkhorn subproblems. In such case, an iteration complexity should be given to justify better time complexity compared to $W_p$. If $p=2$, the proposed Algorithm 2 is almost identical to the Sinkhorn algorithm with an additional step to pre-compute the mean vectors to avoid numeric instability. I don't see much computational advantage in practice for using $RW_p$.

In practice, we need to choose $p$. Different $p$'s seem to give very different results (e.g., Section 5.2). What is the practical and/or theoretical guidance of choosing $p$? Moreover, for $p \neq 2$, how to determine the step size parameter $\eta_1$ in the horizontal gradient descent? Is it constant step size or is it annealed? Any theoretical justification should be helpful to guide the choice of $\eta_1$.

The simulation example in Section 5.1 is too simple. For Gaussians, it is not necessary to compute Wasserstein distances (vanilla and proposed) based on Sinkhorn. One should have closed form for both distances. More extensive settings (such as much higher-dimensional non-Gaussian distributions for general $p \geq 1$) should be used to demonstrate numerical benefit of debiasing the centers.

**Questions:**

Some minor comments:

In Algorithm 1, where was $\eta_2$ used?

In Theorem 4, $\bar{\mu}$ and $\bar{\nu}$ are not defined. They are only defined later.

---

> ### Author Response · Authors · 2024-11-26
>
> Thank you for your valuable feedback and constructive suggestions. Below, we address the weaknesses and questions you raised:
>
> 1. We acknowledge your concern regarding the time complexity claim in Remark 1. While it is true that solving $RW_p$ may require multiple Sinkhorn subproblems, we argue that ${RW}_p$ offers greater robustness compared to $W_p$, especially in datasets with noise or variability. While the computational complexity of ${RW}_p$ may be higher in certain cases, the additional cost is justified by the robustness and adaptability it provides. This makes the method particularly valuable in scenarios where $W_p$ might fail to capture meaningful relationships due to instability or bias. We will clarify these points in the revised manuscript.
>
> 2. In Section 3.2, we discuss the relationship between the ${RW}_p$ metric and the $L_p$ norm. Specifically:
> ${RW}_1$ is robust to noisy datasets, making it well-suited for scenarios with high variability.
> ${RW}_2$ is better suited for cleaner datasets, offering a balance between computational feasibility and precision.
> Higher-order ${RW}_p$ values apply stronger penalties to larger distances, making them useful for applications requiring stricter enforcement of outlier effects.
>
> 3. We acknowledge the simplicity of the example in Section 5.1 and agree that for Gaussian distributions, closed-form solutions for Wasserstein distances are available. To better showcase the utility of our method, we have included additional experiments involving non-Gaussian distributions in D.1. These examples demonstrate the numerical benefits of debiasing the centers and the robustness of ${RW}_p$ in more complex settings.
>
> Thank you for pointing out the use of $\eta$ and the notation of $\bar{\mu}$ and $\bar{\nu}$. We will remove $\eta_2$ from Algorithm 1 for consistency. Additionally, the definitions of $\bar{\mu}$ and $\bar{\nu}$ are already provided in the notations paragraph in Section 1. We will cross-reference this section in the relevant parts of the manuscript to ensure clarity for readers.
>
> Thank you again for your constructive feedback and suggestions. These insights will help improve the clarity, rigor, and practical relevance of our manuscript. Please let us know if further clarification or additional experiments would be helpful.

---

### Official Review · Reviewer_Y95f · 2024-11-02

**Soundness:** 3
**Presentation:** 3
**Contribution:** 1
**Rating:** 3
**Confidence:** 3

**Summary:**

This paper focuses on finding a distance metric between distributions that is invariant to translation while also possessing the desirable properties of the Wasserstein distance. The authors propose a new metric and provide both theoretical and numerical proofs demonstrating that it satisfies the properties of a metric and is invariant to translation. Their analysis is conducted specifically for discrete distributions. Additionally, the paper introduces an algorithm for measuring a special case referred to as RW_2.

**Strengths:**

The paper is well written, and the mathematical proofs are rigorous.

**Weaknesses:**

The literature review section needs improvement; it fails to explain why this problem is important and does not address previous efforts made to solve it. Additionally, the work is not clearly positioned within the existing literature.

**Questions:**

The first approach that comes to mind for solving this problem is to subtract the mean of both distributions, thereby transforming them into zero-mean distributions. We could then calculate the Wasserstein distance between these two zero-mean distributions. However, I am concerned about why the method proposed by the authors is superior to this straightforward option. If the authors can address this question, I may reconsider my score.

---

> ### Author Response · Authors · 2024-11-20
>
> Thanks for your efforts for reviewing the paper.
>
> The straightforward option may not be applicable in all cases. We provide a counterexample in Section C.3 to illustrate that this does not hold for the $RW_1$ metric. Please check the supplementary materials for more information.

---

> > ### Comment · Reviewer_Y95f · 2024-11-21
> >
> > Thank you for your response. I understand that the method yields different results for other norms, such as norm 1. However, the primary motivation of the paper—identifying similar weather patterns—relies on norm 2. In this context, it appears that your method does not demonstrate any advantages over the straightforward method.

---

### Official Review · Reviewer_AePN · 2024-11-03

**Soundness:** 2
**Presentation:** 3
**Contribution:** 3
**Rating:** 5
**Confidence:** 4

**Summary:**

The paper introduces a shifting invariant Wasserstein-based metric called relative-translation invariant Wasserstein distances $RW_p$ to measure the similarity between shifted distributions. We can see this distance generalized the original Wasserstein by being invariant to distribution translations. For $p = 2 $, the $RW_2$ distance shows promising properties such as the decomposability of the optimal transport model and translation invariance of coupling solutions. The authors propose two algorithms for computing general $RW_p$ and a variant of the Sinkhorn algorithm for $RW_2$ computation. Also, the theoretical analysis of numerical stability and time complexity is provided. In the end, they conducted two experiments in comparison with Sinkhorn to validate the new metric performance, which includes a computational time comparison and an image retrieval task with a weather dataset.

**Strengths:**

- Novelty, paper presents a novel shifting-invariant Wasserstein-based metric, which extends the original Wasserstein distance to be invariant under shifting translations
- Clarity and Mathematical Soundness, this paper is generally well-written with clear explanations. The authors provide solid mathematical proofs for the metric properties and the relation to Wasserstein distance.
- Algorithm Development, this paper proposed two algorithm implementations for the $RW_p$ distances, the $RW_2$ Sinkhorn preforms better than the original Sinkhorn according to the experiment results

**Weaknesses:**

- This paper does not provide an analysis of the convergence rates of the $RW_p$ distances as a distribution measure.

- This paper lacks theoretical and experimental comparisons with Gromov-Wasserstein (GW) distance that has similar invariance properties. The GW distance is also translation-invariant and compares distributions based on the shapes, which makes it a good benchmark for comparison in the experiment section [1]. A recent work [2] proposed a robust p-Wasserstein distance (RPW), that claims robustness under shifting perturbations, especially when $p = 2$. A similar image retrieval task was conducted in this work. In general, I feel like the paper lacks comparisons with related works, such as GW distance and RPW both in theory and practice.

With these two major drawbacks, I tend to reject this paper for major revisions.

[1] Mémoli, Facundo. "Gromov–Wasserstein distances and the metric approach to object matching." Foundations of computational mathematics.

[2] Raghvendra, Sharath, Pouyan Shirzadian, and Kaiyi Zhang. "A New Robust Partial p-Wasserstein-Based Metric for Comparing Distributions." Forty-first International Conference on Machine Learning.

**Questions:**

- How does the proposed $RW_p$ distance compare with the GW distance in theory and experiment, which is also translation-invariant and focuses on shape similarity? Which types of distributions translation where the $RW_p$ distances may not perform well compared to other invariant metrics? Again, including such comparisons supports the paper with the broader range of translation invariant Wasserstein-based distances.

- Could you provide theoretical analysis or empirical observations on the convergence rates of $RW_p$?

- Can the $RW_p$ be extended to handle other types of distribution shifts, such as scaling or rotation?

- For the pattern detection experiment, is the thunderstorm snapshot data labeled? If it is, could you provide the accuracy of retrieved similar patterns to better quantify the performance of $RW_p$?

- Is the code public for the experiment section?

---

> ### Author Response · Authors · 2024-11-26
>
> Thank you for your valuable feedback and constructive suggestions. Below, we address the weaknesses and questions you raised:
>
> 1. The GW problem is a highly nonconvex optimization problem and is particularly sensitive to the choice of the initial point. Due to this sensitivity and the fundamental differences in their problem formulations, we believe that a direct comparison with our proposed method may not yield meaningful insights. However, we acknowledge the relevance of GW as a benchmark and will consider further clarifying these distinctions in the manuscript.
>
> 2. The convergence rate of the proposed method matches that of the original Sinkhorn algorithm, as established by Theorem 4.1 on page 70 of reference [1]. Specifically, the convergence depends on the contraction factor $\lambda(K)$, which is determined by $\eta(K)$. For a translation vector $s$ and the translated cost matrix $K'$, we can show that $\eta(K) = \eta(K')$. This follows from the fact that:
> $|| x_i - y_j + s ||_2^2 + || x_k - y_l + s||_2^2 - || x_i - y_l + s ||_2^2 - || x_k - y_j + s||_2^2  = || x_i - y_j  ||_2^2 + || x_k - y_l  ||_2^2 - || x_i - y_l ||_2^2  -  || x_k - y_j ||_2^2.$ Thus, $\eta(K) = \eta(K')$, and the convergence rate remains unaffected by the translation $s$. Therefore, the proposed method's convergence rate is equivalent to that of the original Sinkhorn method, regardless of the translation.
>
> 3. As mentioned in the paper (line 252), the $RW_p$ metric is not designed to handle rotations, where it is possible to find a counterexample again rotation-invariant.
>
> 4. The thunderstorm snapshot data used in the pattern detection experiment is not labeled. Consequently, we cannot provide accuracy metrics for the retrieved patterns. However, we agree that labeled datasets could provide more quantitative evaluations in future experiments.
>
> 5. The code for the experiment section is publicly available. We will ensure this information is explicitly stated in the manuscript for ease of access.
>
> Thank you again for your constructive feedback and suggestions. These insights will help improve the clarity, rigor, and practical relevance of our manuscript. Please let us know if further clarification or additional experiments would be helpful.
>
> [1] Peyré and Cuturi, computational optimal transport, 2019

---

### Official Review · Reviewer_uCLn · 2024-11-04

**Soundness:** 3
**Presentation:** 3
**Contribution:** 2
**Rating:** 5
**Confidence:** 3

**Summary:**

This paper presents a new Wasserstein-based distance function, called the relative-translation invariant Wasserstein distance $RW_p$, which given two distributions $\mu$ and $\nu$, finds the optimal pair of shift and transport map $(s, P)$ that minimizes the transport cost of $P$ between $\mu$ and the translated distribution $\nu$ by a vector $s$. It is shown that this distance is a metric and can be computed by computing the gradients and alternatively updating the shift $s$ and the transport map $P$.

When $p=2$, the paper shows that the object function of $RW_2$ can be expressed as minimizing the sum of two functions, where one is independent of the shift and the other is independent of the transport map. In this case, the authors use the Sinkhorn algorithm to speed up the computation of $RW_2$. Furthermore, the authors show that the diameter of the point sets after applying the optimal shift would reduce for empirical distributions derived from sub-Gaussian distributions, hence improving the execution time of the Sinkhorn algorithm.

**Strengths:**

- The paper presents the properties of finding the Wasserstein distance when we are allowed to shift the distributions.
- The paper also presents algorithms for approximating the $RW_p$.

**Weaknesses:**

- I think the proof of triangle inequality has some mistakes. My main concern is where you set $W_p(\eta, \eta')$ to 0 from line 2 to line 3. If $[W_p]$ refers to the Wasserstein distance between the classes of $\mu$ and $\nu$, then essentially $RW_p=[W_p]$ and from line 1 to line 2 of the Equation, you are assuming the triangle inequality holds for your distance. If $[W_p]$ just means the Wasserstein distance, then $W_p(\eta, \eta')$ might not be 0. (It seems reasonable that $RW_p$ is a metric, so my concern mostly is that the proof is not written correctly and not that the theorem is incorrect.)
- It is good that the method is implementable and can be used to test the performance of the new distance function, but current experimental results do not convey the message. For the numerical validation part, see my questions below. For the thunderstorm pattern detection, although visually $RW_2$ performs better than $W_2$ in Figure 4, I would say it is hard to judge the results in Figure 9 and Figure 8. I would strongly suggest adding experiments with a ground truth to your next version; something like an image retrieval task on labeled images, where you randomly shift a dataset of labeled images and show an improvement in the accuracy of the retrieved images for each query image when using $RW_2$ instead of $W_2$.

**Questions:**

- Corollary 1 seems to be extendable to all values of p and not only p=2. The proof of it might not be straightforward, but I am curious to know if you already have some counter-examples showing that if $P$ is an optimal map for the untranslated distributions, then after translating the distributions, the same $P$ would not have an optimal cost.
- For the experiments on numerical validation, can you explain why the Sinkhorn algorithm running time is not changing for the values of $s\in[0,2.4]$ and then suddenly drops to around 0 times (Figure 3 (c))? Based on your discussion in Section 4.4, when the shift increases, the diameter should also increase and we should expect to see higher running times.
- In the same experiment, why is there a sharp increase in the error of the Sinkhorn algorithm for $s>2.4$ (Figure 3 (d))? Am I right that for $s>2.4$, the Sinkhorn algorithm did not compute anything and just returned 0 as the output cost?
- Figure 3 (b): what is the computation error? Why do the two methods have the same set of errors for all values of s?
- Also, based on the experiments in the appendix on numerical validation, there are sharp increases and decreases in the results, which is unintuitive. What is the number of runs you performed for each value of $s$ that resulted in those plots?

---

> ### Author Response · Authors · 2024-11-20
>
> Thanks for your efforts for reviewing the paper.
>
> For the questions:
> 1. The reviewer’s straightforward approach may not apply universally, particularly in cases such as \( p = 1 \). A counterexample is provided in Section C.3, where two distributions are used to demonstrate that the optimal solution is not invariant to translation when the metric is $RW_1$. Please refer to the supplementary materials for further details.
>
> 2. This is an excellent question. The primary reason for the dropout near zero is that as $s$ increases, it leads to division-by-zero issues, causing significant numerical errors, however, which also make the program faster to compute. This results in faster convergence and reduced runtime. You may check more details by testing the uploaded files.
>
> 3. This is the same as point 2.
>
> 4. The computational error is calculated as the difference between the Wasserstein-2 distances obtained using the Sinkhorn/$RW_2$-Sinkhorn method and the true Wasserstein-2 distances computed via linear programming. As shown in Figure 3, the errors are nearly identical, with slight variations around $1 \times 10^{-5} $.
>
> 5. The unstable images are also a result of division-by-zero issues, as previously mentioned, demonstrating that the $RW_2$-Sinkhorn method is more stable than the regular Sinkhorn approach.

---

> > ### Comment · Reviewer_uCLn · 2024-11-20
> >
> > 1- I am not sure if the counter example addresses my question. I want to re-state my question: Consider distributions $\mu$ and $\nu$ and an optimal map $P$ between $\mu$ and $\nu$. Now, let $\mu'$ be a translation of $\mu$ corresponding to the best translation. Then, can we say that the same transport map $P$ is a minimum cost transport map between $\mu$ and $\nu$? In the example in C.3, the matching $(x_1, y_2), (x_2, y_1), (x_3, y_3)$ is optimal both before and after translation, is that correct?
> >
> > 2,3- I believe as you said, the execution of Sinkhorn lead to an exception of division by zero, which made your program to stop much earlier and return 0 as the cost. That is why you see the sudden drop of time to 0 and the sudden jump of error to high values. Can you provide the cost of the transport plans computed by Sinkhorn and RW2 for that experiment?
> >
> > 4- In your experiments, you shift one distribution by a vector s; this translation should pose some error in the Sinkhorn computation that is not present in your distance, right? So why do they have the same cost? In fact, your method should return a value close to 0, whereas Sinkhorn should return a value around |s|.

---

### Official Review · Reviewer_1638 · 2024-11-04

**Soundness:** 2
**Presentation:** 3
**Contribution:** 2
**Rating:** 3
**Confidence:** 4

**Summary:**

Many machine learning pipelines rely on training objective losses involving comparing probabilities measures. One of the mostly used in these pipelines are optimal transport (OT), aka Wasserstein distance,  that leverages the geometrical information of the distributions in question. In a nutshell, Wasserstein distance seeks for the cheaper cost to transport a source distribution to a target one, where the optimization problem behind consists in a linear programming problem.

This paper addresses a major limitation of classical Wasserstein distance that corresponds to a translation shift in the source distribution. It introduces a family of relative-translation invariant Wasserstein $RW_p$ that behave like to Wasserstein distance and invariant to translation. For $p=2$, $RW_2$ enjoys a decomposition of the relative translation optimal transport (ROT). In addition, $RW_2$ can be solved using Sinkhorn iterations. The papers ends up with empirical results on weather detection dataset.

**Strengths:**

- The authors propose relative translation optimal transport (ROT) which induces a Wasserstein like distance on the quotient space of probability measures induced by the translation relation.
- For quadratic ROT, a decomposition in terms of Wasserstein distance (horizontal function) summed with a quadratic term including the shifting parameter $s$.
- The authors test ROT approach on weather detection dataset.
- I checked the proofs of the main results and they sound correct.

**Weaknesses:**

- In Definition 4, the relative transplantation invariant Wasserstein, $RW_p(\mu, \nu)$ is given with respect to an $s$-shifting of the source distribution $\mu$. However in Theorem 2, $RW_p$ is proper distance on the quotient set of shifting probabilities. Since the main results of the paper are based with respect to a shifting of the source, I’am wondering about the properties of $RW_p$ outside the quotient set.
- The quadratic $RW_2$ is very closed to the translation property of Wasserstein distance with a quadratic cost (see Remark 2.19 in Peyré and Cuturi, 2019). namely, if one considers only the case of shifting the source distribution, the decomposition of Quadratic ROT can be straiforwardly from this remark and  a simple optimization minimisation over the shifting parameter $s$. I think this point weakens the novelty of this paper.
- I think that the computational efficiency of Sinkhorn $RW_2$ over classical Sinkhorn is not significant since the maximum over the shift cost matrix needs a lower bound of the L2 norm $\||\bar{\mu} - \bar{\mu}\||$ with an order of $\sqrt{n}$.

**Questions:**

**Minor Typos**
- M179: « coupling «  —> « Coupling »
- L325: « Thoorem »

---

> ### Author Response · Authors · 2024-11-26
>
> Thank you for your valuable feedback. Below, we address the points raised in your review:
>
> For the weaknesses:
> Thank you for highlighting this question. Could you please clarify what specific properties you are referring to outside the quotient set? While the main results focus on the behavior within the quotient set, further exploration of properties beyond this set may require additional assumptions or analysis. We would appreciate a more detailed explanation to better address your concerns.
>
> We acknowledge your observation regarding the connection to Remark 2.19 in Peyré and Cuturi (2019). While the decomposition of Quadratic ROT for shifting distributions aligns with this observation, our contribution focuses on leveraging this decomposition to extend the analysis specifically for the case of the relative transplantation invariant Wasserstein distance. The cited reference is mentioned in the subsequent paragraph of Corollary 2. We can consider removing this part or reframing it to emphasize how our approach offers a complementary perspective, thereby enriching the understanding of the decomposition.
>
> Regarding your comment on the order of $\sqrt{n}$, it is not entirely clear what aspect of the computational complexity you are referencing. If your comment pertains to the dependency on the $\ell_2$-norm or the shifting cost matrix, could you provide more details? Our experimental results demonstrate computational advantages of the proposed Sinkhorn variant, particularly in scenarios where the shifting parameter is bounded. A clearer explanation will help us refine and strengthen our arguments in this context.
>
> Thank you for pointing out the typographical errors. We will make the corrections.
>
> We appreciate your feedback and suggestions, as they will help improve the clarity and rigor of our work. Please let us know if further clarification is needed on any of the points raised.

---

### Public Comment · ~Zhiwei_Jia1 · 2024-11-17
**How about translation between distributions with unmatched semantics statistics?**

Hi authors,

Thanks for the work. I wonder in a generalized case of simple translation (distribution shift), where two distributions differ by also in the underlying semantic statistics (see discussion in [1]), for instance, for domain adaptation tasks using unpaired image sets, will the proposed metric help?

[1] Semantically Robust Unpaired Image Translation for Data With Unmatched Semantics Statistics, ICCV 2021

---

### Meta-Review · Area_Chair_GWFh · 2024-12-17

**Metareview:**

The authors propose a novel relative-translation invariant Wasserstein ($RW_p$) distance to deal with the translation shift for Wasserstein distance. The authors prove that it satisfies the metric property. The authors propose two algorithmic approach for the proposed problem: two-stage optimization, and a variant of Sinkhorn algorithm. The authors test the proposed distance on a toy example and thunderstorm pattern detection task.

The Reviewers raised concerns on the related works/baselines for the translation shift with optimal transport based approach (e.g., Gromov Wasserstein and robust partial p-Wasserstein which are highly relevant to the considered problem), and the simple baseline based on zero-means (which is highly related to the case $p=2$, and the translation property of the standard optimal transport with squared Euclidean cost). The Reviewers raised concerns about the instability (e.g., dividing by zero issue in which more attention should be required to address). The Reviewers also raised concerns on the empirical evidence for the advantages of the proposed method (e.g., the toy example with Gaussian distribution is not clear, given the closed-form solution of Wasserstein; and there is no ground truth on the thunderstorm patter detection task which leads to unclear evaluation on advantages of the proposed approach. Additionally, it is not clear for the Reviewers why the proposed method can have better time complexity than the standard OT given its definition.

Overall, it is necessary for a major revision to improve the submission. The authors may consider the Reviewers' comments to revise the submission.

**Additional Comments On Reviewer Discussion:**

The raised points are given in the meta-review which are not convincingly addressed yet from the rebuttal. Therefore, a major revision is required to address the raised concerns.

---

### Decision · Program_Chairs · 2025-01-22

Reject